# Asynchronous Policy Gradient Aggregation for Efficient Distributed Reinforcement Learning

**Alexander Tyurin**
AXXX, Moscow, Russia
Applied AI Institute, Moscow, Russia

**Andrei Spiridonov**
AXXX, Moscow, Russia

**Varvara Rudenko**
AXXX, Moscow, Russia

## Abstract

We study distributed reinforcement learning (RL) with policy gradient methods under *asynchronous and parallel computations and communications*. While non-distributed methods are well understood theoretically and have achieved remarkable empirical success, their distributed counterparts remain less explored, particularly in the presence of heterogeneous asynchronous computations and communication bottlenecks. We introduce two new algorithms, Rennala NIGT and Malenia NIGT, which implement asynchronous policy gradient aggregation and achieve state-of-the-art efficiency. In the homogeneous setting, Rennala NIGT provably improves the total computational and communication complexity while supporting the AllReduce operation. In the heterogeneous setting, Malenia NIGT simultaneously handles asynchronous computations and heterogeneous environments with strictly better theoretical guarantees. Our results are further corroborated by experiments, showing that our methods significantly outperform prior approaches.

## 1 Introduction

Reinforcement Learning (RL) is one of the most important tools in modern deep learning and large language model training. There are many applications, including robotics and control (Kober et al., 2013; Levine et al., 2016), recommender systems (Chen et al., 2019), and game playing such as Go, chess, and StarCraft II (Silver et al., 2017; Vinyals et al., 2019). Moreover, RL has used in the fine-tuning of large language models through reinforcement learning from human feedback (RLHF) (Christiano et al., 2017).

Modern machine learning and RL models are typically trained in a distributed fashion, where many agents, workers, or GPUs perform asynchronous computations in parallel and communicate periodically (Dean et al., 2012; Recht et al., 2011; Goyal et al., 2017). This distributed paradigm enables training at scale by efficiently utilizing massive computational resources. However, distributed training poses many challenges, including communication bottlenecks (Alistarh et al., 2017; Lin et al., 2017), system heterogeneity across agents (Lian et al., 2015), and stragglers and fault tolerance (Chen et al., 2016). These challenges have motivated a large body of work on communication-efficient methods, asynchronous and decentralized optimization algorithms, and federated learning frameworks (Konečný et al., 2016; McMahan et al., 2017; Kairouz et al., 2021).

Policy Gradient (PG) methods are among the most popular and effective classes of RL algorithms (Williams, 1992; Sutton et al., 1998). They have demonstrated remarkable empirical performance across a wide range of challenging tasks. In non-distributed settings, significant recent progress has been made, and PG-based methods are now well studied and better understood (Yuan et al., 2022; Masiha et al., 2022; Ding et al., 2022; Lan, 2023; Fatkhullin et al., 2023). However, distributed RL scenarios are less studied, and many critical challenges and open questions remain.

In this work, we study state-of-the-art policy-based methods in *parallel and asynchronous* setups, where a large number of agents collaborate to maximize the expected return of a discrete-time discounted Markov decision process. Our focus is on addressing *computational and communication challenges* that arise in distributed RL.

## 1.1 RELATED WORK

**Non-distributed PG methods.** In tabular settings, Lan (2023) study policy mirror descent and establish global guarantees for discrete state–action spaces. Alfano & Rebeschini (2022); Yuan et al. (2023) analyze PG methods with log-linear parameterizations. Our focus, however, is on general continuous state–action spaces with trajectory sampling and stochastic policy gradients. In this setting, Yuan et al. (2022) analyze Vanilla PG and prove a sample complexity of $\mathcal{O}(\varepsilon^{-4})$ for finding an $\varepsilon$-stationary point. Several PG variants have since improved this rate. In particular, Huang et al. (2020); Ding et al. (2022); Xu et al. (2020a;b); Fan et al. (2021) introduce momentum-based PG methods, but their analyses rely on importance sampling (IS) and therefore require strong additional assumptions (e.g., variance of IS weights is bounded, as in Assumption 4.4 of (Xu et al., 2020a)), which we ideally want to avoid. More recently, Fatkhullin et al. (2023), based on (Cutkosky & Mehta, 2020), propose a normalized PG method that improves the rate for finding an $\varepsilon$-stationary point to $\mathcal{O}(\varepsilon^{-7/2})$ under much weaker assumptions, requiring only second-order smoothness (without explicit use of Hessians). Hessian-aided PG methods have also been investigated (Fatkhullin et al., 2023; Ganesh et al., 2024); however, in practice, the stochastic Hessian–vector products sampled in these methods have much higher variance than stochastic gradients (Fatkhullin et al., 2023).

**Parallel and asynchronous optimization.** Distributed optimization is typically considered in two settings: homogeneous and heterogeneous. Both settings are equally important. The former, in the context of RL, implies that all agents access the same environment and data, while the heterogeneous setting, which is more prevalent in federated learning (FL) (Konečný et al., 2016), arises when the data and environments differ due to privacy constraints or the infeasibility of sharing environments.

In the *homogeneous* case, many studies have analyzed asynchronous variants of stochastic gradient descent, such as (Lian et al., 2015; Feyzmahdavian et al., 2016; Stich & Karimireddy, 2020; Sra et al., 2016). A common limitation of these works is the requirement that the delays in stochastic gradient indices remain bounded. Consequently, their results provide weaker guarantees on computational time complexity compared to more recent analyses (Cohen et al., 2021; Koloskova et al., 2022; Mishchenko et al., 2022; Tyurin & Richtárik, 2023; Maranjyan et al., 2025), which avoid this restriction. In the *heterogeneous* case, many works have also been proposed, including (Mishchenko et al., 2022; Koloskova et al., 2022; Wu et al., 2022; Tyurin & Richtárik, 2023; Islamov et al., 2024).

In order to compare parallel methods, Mishchenko et al. (2022) proposed the $h_i$-*fixed computation model* in the context of stochastic optimization (see Assumption 2.5 in the context of RL), assuming that it takes at most $h_i$ seconds to calculate one stochastic gradient on agent $i$. Mishchenko et al. (2022); Koloskova et al. (2022) provided new analyses and proofs of Asynchronous SGD, showing that their versions of Asynchronous SGD have the time complexity $\mathcal{O}\big((1/n \sum_{i=1}^{n} 1/h_i)^{-1}(1/\varepsilon + 1/n\varepsilon^2)\big)$, where the Big-$\mathcal{O}$ notation is up to an error tolerance $\varepsilon$ and $h_i$. Surprisingly, this complexity can be improved[1] to $\Theta(\min_{m \in [n]}[(1/m \sum_{i=1}^{m} 1/h_i)^{-1}(1/\varepsilon + 1/m\varepsilon^2)])$, achieved by the Rennala SGD method from (Tyurin & Richtárik, 2023). For the heterogeneous setting, they also developed the Malenia SGD method with the time complexity $\Theta(\max_{i \in [n]} h_i/\varepsilon + (1/n \sum_{i=1}^{n} 1/h_i)1/n\varepsilon^2)$. Moreover, Tyurin & Richtárik (2023) proved that these complexities are optimal under smoothness and the bounded stochastic gradient assumption.

In the distributed RL domain, numerous works have been proposed. For instance, Fan et al. (2021); Ganesh et al. (2024) analyzed federated reinforcement learning with fault-tolerance approaches in settings with adversarial attacks, Jin et al. (2022); Ganesh et al. (2024); Labbi et al. (2025) studied federated policy gradient in synchronous homogeneous and heterogeneous settings, while Lu et al. (2021); Chen et al. (2021) investigated decentralized policy optimization with a focus on communication efficiency.

**The current state-of-the-art method in asynchronous and parallel RL.** The most relevant and central work for our study is the recent result of Lan et al. (2025), which considers an asynchronous PG method called AFedPG and provides the current state-of-the-art time complexity under our assumptions. Their method builds on the idea of applying the NIGT method (Cutkosky & Mehta, 2020; Fatkhullin et al., 2023) to the asynchronous RL domain, aided by recent insights from Koloskova et al. (2022); Mishchenko et al. (2022). It is worth noting that this combination is not straightforward and requires several technical steps to adapt the previous analysis to RL problems.

---

[1] Notice that $\min_{m \in [n]} g(m) \le g(n)$ for any $g : \mathbb{N} \to \mathbb{R}$.

Table 1: **Homogeneous Setup.** The time complexities of distributed methods to find an $\varepsilon$-stationary point in problem (1) up to an error tolerance $\varepsilon$, number of agents $n$, computation times $\dot{h}_i$, communication time $\kappa$ (see Section 4.1), and ignoring logarithmic factors.

| Method | Total Time Complexity | | | Support AllReduce |
|---|---|---|---|---|
| | **Computational Complexity** | | **Communication Complexity** | |
| (Syncronous) Vanilla PG (Yuan et al., 2022) | $\max\limits_{i\in[n]} \dot{h}_i \left( \frac{1}{\varepsilon^2} + \frac{1}{n\varepsilon^4} \right)$ | $+$ | $\kappa \times \left( \frac{1}{\varepsilon^2} + \frac{1}{n\varepsilon^4} \right)$ | **Yes** |
| (Syncronous) NIGT (Fatkhullin et al., 2023) | $\Omega\left( \max\limits_{i\in[n]} \dot{h}_i \times \frac{1}{n\varepsilon^{7/2}} \right)$ | $+$ | $\kappa \times \frac{1}{\varepsilon^{7/2}}$ | **Yes** |
| Rennala PG/SGD (Tyurin & Richtárik, 2023) | $\min\limits_{m\in[n]} \left[ \left( \frac{1}{m}\sum\limits_{i=1}^{m} \frac{1}{\dot{h}_i} \right)^{-1} \left( \frac{1}{\varepsilon^2} + \frac{1}{m\varepsilon^4} \right) \right]$ | $+$ | $\kappa \times \frac{1}{\varepsilon^2}$ | **Yes** |
| AFedPG (Lan et al., 2025) | $\left( \frac{1}{n}\sum\limits_{i=1}^{n} \frac{1}{\dot{h}_i} \right)^{-1} \left( \frac{n^{4/3}}{\varepsilon^{7/3}} + \frac{1}{n\varepsilon^{7/2}} \right)$ | $+$ | $\kappa \times \frac{1}{\varepsilon^3}$ | No |
| Rennala NIGT (**new**) (Theorem 4.4) | $\min\limits_{m\in[n]} \left[ \left( \frac{1}{m}\sum\limits_{i=1}^{m} \frac{1}{\dot{h}_i} \right)^{-1} \left( \frac{1}{\varepsilon^2} + \frac{1}{m\varepsilon^{7/2}} \right) \right]$ | $+$ | $\kappa \times \frac{1}{\varepsilon^2}$ | **Yes** |
| Lower bound[a] (**new**) (Theorem G.1) | $\min\limits_{m\in[n]} \left[ \left( \frac{1}{m}\sum\limits_{i=1}^{m} \frac{1}{\dot{h}_i} \right)^{-1} \left( \frac{1}{\varepsilon^{3/2}} + \frac{1}{m\varepsilon^{7/2}} \right) \right]$ | $+$ | $\kappa \times \frac{1}{\varepsilon^{12/7}}$ | — |

**The total time complexity of** Rennala NIGT **is better than that of previous methods:**
i) for small-$\varepsilon$, its *computational complexity* is much better than that of Vanilla PG, NIGT, and Rennala PG;
ii) Rennala NIGT supports AllReduce and its *communication complexity* is much better than that of AFedPG for small-$\varepsilon$;
iii) its *computational complexity* can be arbitrarily better[2] than that of AFedPG, whose complexity can grow with $n$.

[a] The lower bound applies to methods that only access unbiased stochastic gradients of an $(L_g, L_h)$–twice smooth function with $\sigma$–bounded variance, treating (3) as a black-box oracle. Extending it to methods exploiting the full structure of $J$ in (1) and closing the gap remains open. See discussion in Sections 4.1 and G.

However, AFedPG has several limitations: i) although motivated by and designed for federated learning, this method does not support the heterogeneous setting; ii) due to its greedy update strategy, the method has *suboptimal communication time complexity* (as we illustrate in Section 4.1 and Table 1) and does not support the AllReduce operation, which is essential in most distributed environments; iii) finally, its *computational time complexity is also suboptimal* and can be improved (Table 1).

## 1.2 CONTRIBUTIONS

We develop two new methods, Rennala NIGT and Malenia NIGT, which achieve new state-of-the-art computational and communication time complexities in the homogeneous and heterogeneous settings, respectively. Our theory strictly improves upon the result of Lan et al. (2025) in all aspects *within the same setting, without requiring additional assumptions*: i) we develop and analyze Malenia NIGT, which supports asynchronous computations in the heterogeneous setup; ii) our *communication time complexity* is provably better in the homogeneous setup. For example, in the small–$\varepsilon$ regime, Rennala NIGT improves AFedPG's bound of $\mathcal{O}(\kappa\varepsilon^{-3})$ to $\mathcal{O}(\kappa\varepsilon^{-2})$. And in the worst case, the communication complexity of AFedPG can provably be as large as $\mathcal{O}(\kappa\varepsilon^{-7/2})$ (see Section 4.1). Furthermore, both Rennala NIGT and Malenia NIGT support AllReduce; iii) our *computational time complexity* is also strictly better in the homogeneous setup. In the small–$\varepsilon$ regime, we improve their complexity $\tilde{\mathcal{O}}\left( (1/n \sum_{i=1}^{n} 1/\dot{h}_i)^{-1} (n^{4/3}/\varepsilon^{7/3} + 1/n\varepsilon^{7/2}) \right)$ to $\tilde{\mathcal{O}}\left( \min_{m\in[n]}[(1/m \sum_{i=1}^{m} 1/\dot{h}_i)^{-1}(1/\varepsilon^2 + 1/m\varepsilon^{7/2})] \right)$, which can be arbitrarily smaller (see the discussion in Section 4). Even in the classical optimization setting, we significantly improve upon the current state-of-the-art results of Tyurin & Richtárik (2023); Maranjyan et al. (2025) by leveraging momentum and normalization techniques, together with a mild second-order smoothness assumption. As a final contribution, we establish a new lower bound, Theorem G.1 from Section G, which enables us to quantify the remaining optimality gap.

We believe our new methods, analysis, insights, and numerical experiments are important for the RL community, as they achieve state-of-the-art time complexities in asynchronous and parallel RL (Tables 1 and 2), an area that is rapidly gaining importance with the growing availability of computational resources.

---

[2] $\min\limits_{m\in[n]} \left[ \left( \frac{1}{m}\sum\limits_{i=1}^{m} \frac{1}{\dot{h}_i} \right)^{-1} \left( \frac{1}{\varepsilon^2} + \frac{1}{m\varepsilon^{7/2}} \right) \right] \leq \left( \frac{1}{n}\sum\limits_{i=1}^{n} \frac{1}{\dot{h}_i} \right)^{-1} \left( \frac{1}{\varepsilon^2} + \frac{1}{n\varepsilon^{7/2}} \right) \leq \left( \frac{1}{n}\sum\limits_{i=1}^{n} \frac{1}{\dot{h}_i} \right)^{-1} \left( \frac{n^{4/3}}{\varepsilon^{7/3}} + \frac{1}{n\varepsilon^{7/2}} \right)$

## 2 PROBLEM FORMULATION AND PRELIMINARIES

We consider a discrete-time discounted Markov decision process $\mathcal{M} = (\mathcal{S}, \mathcal{A}, \mathcal{P}, r, \rho, \gamma)$, where $\mathcal{S}$ and $\mathcal{A}$ denote the state and action spaces, $\mathcal{P} : \mathcal{S} \times \mathcal{A} \times \mathcal{S} \to \mathbb{R}$ is the transition kernel, $r : \mathcal{S} \times \mathcal{A} \to [-r_{\max}, r_{\max}]$ is the reward function bounded by $r_{\max} > 0$, $\rho$ is the initial state distribution, and $\gamma \in (0, 1)$ is the discount factor (Puterman, 2014).

We assume $n$ agents operate asynchronously in parallel and interact with independent copies of the environment. Each agent selects an action $a_t \in \mathcal{A}$ in a state $s_t \in \mathcal{S}$ according to the parameterized density function $\pi_\theta(\cdot|s_t)$. The agent then receives the reward $r(s_t, a_t)$, and the environment transitions to the next state $s_{t+1}$ according to the distribution $\mathcal{P}(\cdot|s_t, a_t)$. One of the main problems in RL is to find $\theta \in \mathbb{R}^d$ that maximize the expected return

$$\max_{\theta \in \mathbb{R}^d} \left\{ J(\theta) = \mathbb{E}_{(s_t, a_t)_{t \geq 0}} \left[ \sum_{t=0}^{\infty} \gamma^t r(s_t, a_t) \right] \right\}, \tag{1}$$

where $s_0 \sim \rho(\cdot)$, $a_t \sim \pi_\theta(\cdot|s_t)$, and $s_{t+1} \sim \mathcal{P}(\cdot|s_t, a_t)$ for all $t \geq 0$. In practice, it is infeasible to generate an infinite trajectory. Instead, each agent generates a finite trajectory $\tau = (s_0, a_0, \cdots, s_{H-1}, a_{H-1})$ of length $H \geq 1$ from the density $p(\tau|\pi_\theta) := \rho(s_0)\pi_\theta(a_0|s_0) \prod_{t=1}^{H-1} \mathcal{P}(s_t|s_{t-1}, a_{t-1})\pi_\theta(a_t|s_t)$. We define the truncated expected return:

$$J_H(\theta) = \mathbb{E}_\tau \left[ \sum_{t=0}^{H-1} \gamma^t r(s_t, a_t) \right], \tag{2}$$

which approximates $J(\theta)$. Given the rewards, we can compute the truncated stochastic policy gradient

$$g_H(\tau, \theta) = \sum_{t=0}^{H-1} \left( \sum_{h=t}^{H-1} \gamma^h r(s_h, a_h) \right) \nabla \log \pi_\theta(a_t|s_t), \tag{3}$$

where $\tau = (s_0, a_0, \cdots, s_{H-1}, a_{H-1}) \sim p(\cdot|\pi_\theta)$. The random vector $g_H(\tau, \theta)$ is an unbiased estimator of $\nabla J_H(\theta)$ (Sutton et al., 1998; Masiha et al., 2022): $\nabla J_H(\theta) = \mathbb{E}_\tau[g_H(\tau, \theta)]$. In total, sampling trajectories and calculating $g_H$, the goal of the agents is to find an $\varepsilon$–stationary point $\theta \in \mathbb{R}^d$, i.e., a point $\theta$ such that $\mathbb{E}[\|\nabla J(\theta)\|] \leq \varepsilon$.

### 2.1 ASSUMPTIONS

We consider the standard assumptions from the RL literature:

**Assumption 2.1.** For all $s, a \in \mathcal{S} \times \mathcal{A}$, the function $\theta \to \pi_\theta(a, s)$ is positive, twice continuously differentiable, $\|\nabla \log \pi_\theta(a|s)\| \leq M_g$, and $\|\nabla^2 \log \pi_\theta(a|s)\| \leq M_h$ for all $\theta \in \mathbb{R}^d$, where $M_g, M_h > 0$.

**Assumption 2.2.** For all $s, a \in \mathcal{S} \times \mathcal{A}$, the function $\theta \to \pi_\theta(a|s)$ is positive, twice continuously differentiable, and there exists $l_2 > 0$, such that $\|\nabla^2 \log \pi_\theta(a|s) - \nabla^2 \log \pi_{\bar{\theta}}(a|s)\| \leq l_2 \|\theta - \bar{\theta}\|$ for all $\theta, \bar{\theta} \in \mathbb{R}^d$.

Using the assumptions, we can derive the following useful properties.

**Proposition 2.3** (e.g. (Zhang et al., 2020; Masiha et al., 2022; Yuan et al., 2022)). *Let Assumptions 2.1 and 2.2 hold. Then,*

1. *Function $J$ satisfies $\|\nabla J(\theta) - \nabla J(\theta')\| \leq L_g \|\theta - \theta'\|$ for all $\theta, \theta' \in \mathbb{R}^d$, where $L_g := r_{\max}(M_g^2 + M_h)/(1-\gamma)^2$.*

2. *Function $J$ satisfies $\|\nabla^2 J(\theta) - \nabla^2 J(\theta')\| \leq L_h \|\theta - \theta'\|$ for all $\theta, \theta' \in \mathbb{R}^d$, where $L_h := \frac{r_{max}M_gM_h}{(1-\gamma)^2} + \frac{r_{max}M_g^3(1+\gamma)}{(1-\gamma)^3} + \frac{r_{max}M_g}{1-\gamma} \max\left\{ M_h, \frac{\gamma M_g^2}{1-\gamma}, \frac{l_2}{M_g}, \frac{M_h\gamma}{1-\gamma}, \frac{M_g(1+\gamma)+M_h\gamma(1-\gamma)}{1-\gamma^2} \right\}$.*

3. *$\|\nabla J_H(\theta) - \nabla J(\theta)\| \leq D_g\gamma^H$ and $\|\nabla^2 J_H(\theta) - \nabla^2 J(\theta)\| \leq D_h\gamma^H$ for all $\theta \in \mathbb{R}^d$, where $D_g := \frac{M_g r_{max}}{1-\gamma}\sqrt{\frac{1}{1-\gamma} + H}$ and $D_h := \frac{(M_h+M_g^2)r_{max}}{1-\gamma}\left(\frac{1}{1-\gamma} + H\right)$.*

4. *For $g_H$ defined in (3), we have $\nabla J_H(\theta) = \mathbb{E}_\tau[g_H(\tau, \theta)]$ and $\mathbb{E}_\tau\left[\|g_H(\tau, \theta) - \nabla J_H(\theta)\|^2\right] \leq \sigma^2$ $\forall \theta \in \mathbb{R}^d$ with $\sigma^2 := r_{\max}^2 M_g^2/(1-\gamma)^3$.*

Table 2: **Heterogeneous Setup.** The time complexities of distributed methods to find an $\varepsilon$-stationary point in problem (1) up to an error tolerance $\varepsilon$, computation times $\dot{h}_i$, communication time $\kappa$ (see Section 4.1), and ignoring logarithmic factors.

| Method | Total Time Complexity | | Support AllReduce |
|---|---|---|---|
| | **Computational Complexity** | **Communication Complexity** | |
| (Syncronous) Vanilla PG (Yuan et al., 2022) | $\max\limits_{i \in [n]} \dot{h}_i \frac{1}{\varepsilon^2} + \max\limits_{i \in [n]} \dot{h}_i \frac{1}{n\varepsilon^4}$ + | $\kappa \times \left( \frac{1}{\varepsilon^2} + \frac{1}{n\varepsilon^4} \right)$ | **Yes** |
| Malenia PG/SGD (Tyurin & Richtárik, 2023) | $\max\limits_{i \in [n]} \dot{h}_i \cdot \frac{1}{\varepsilon^2} + \left( \frac{1}{n} \sum\limits_{i=1}^{n} \dot{h}_i \right) \cdot \frac{1}{n\varepsilon^4}$ + | $\kappa \times \frac{1}{\varepsilon^2}$ | **Yes** |
| AFedPG (Lan et al., 2025) | does not support heterogeneous setup | | |
| Malenia NIGT (**new**) (Theorem 5.1) | $\max\limits_{i \in [n]} \dot{h}_i \cdot \frac{1}{\varepsilon^2} + \left( \frac{1}{n} \sum\limits_{i=1}^{n} \dot{h}_i \right) \cdot \frac{1}{n\varepsilon^{7/2}}$ + | $\kappa \times \frac{1}{\varepsilon^2}$ | **Yes** |

**Similar to** Rennala NIGT (**Table 1**), the total time complexity of Malenia NIGT **is better than that of previous methods in the *heterogeneous setup*.**

## 2.2 HOMOGENEOUS AND HETEROGENEOUS SETUPS

We consider two important settings: *homogeneous* and *heterogeneous*. The homogeneous setting arises in open-data scenarios, where each agent has access to the same environment and distribution. In contrast, the heterogeneous setting is more relevant in federated learning (FL) (Konečný et al., 2016; Kairouz et al., 2021), where agents aim to preserve privacy or where sharing environments is infeasible.

**Homogeneous setup.** We start with the homogeneous setup, where agents have access to the same distribution, share the same reward function, and $\pi_\theta$. This problem is the same as in (Lan et al., 2025) and defined in Section 2.

**Heterogeneous setup.** However, unlike (Lan et al., 2025), our theory also supports the heterogeneous setting, where agents have access to arbitrary heterogeneous distributions and reward functions, which is important in FL. We consider the problem of maximizing

$$J(\theta) = \frac{1}{n} \sum_{i=1}^{n} J_i(\theta), \text{ where } J_i(\theta) = \mathbb{E}_{(s_{i,t}, a_{i,t})_{t \geq 0}} \left[ \sum_{t=0}^{\infty} \gamma^t r_i(s_{i,t}, a_{i,t}) \right] \quad (4)$$

and $s_{i,0} \sim \rho_i(\cdot)$, $a_{i,t} \sim \pi_{i,\theta}(\cdot|s_{i,t})$, and $s_{i,t+1} \sim \mathcal{P}_i(\cdot|s_{i,t}, a_{i,t})$ for all $t \geq 0, i \in [n]$. We assume the fully general setting, where agents sample from arbitrary heterogeneous distributions, and both the reward function and $\pi_{i,\theta}$ may differ. The truncated expected return and stochastic gradient are defined as

$$J_H(\theta) = \frac{1}{n} \sum_{i=1}^{n} J_{i,H}(\theta) \text{ and } g_{i,H}(\tau_i, \theta) = \sum_{t=0}^{H-1} \left( \sum_{h=t}^{H-1} \gamma^h r_i(s_{i,h}, a_{i,h}) \right) \nabla \log \pi_{i,\theta}(a_{i,t}|s_{i,t}) \quad (5)$$

where $J_{i,H}(\theta) = \mathbb{E}_{\tau_i} \left[ \sum_{t=0}^{H-1} \gamma^t r_i(s_{i,t}, a_{i,t}) \right]$ and $\tau_i = (s_{i,0}, a_{i,0}, \cdots, s_{i,H-1}, a_{i,H-1})$ from the density $p_i(\tau|\pi_{i,\theta}) := \rho_i(s_0)\pi_{i,\theta}(a_{i,0}|s_{i,0}) \prod_{t=1}^{H-1} \mathcal{P}_i(s_{i,t}|s_{i,t-1}, a_{i,t-1})\pi_{i,\theta}(a_{i,t}|s_{i,t})$. Similarly to Section 2, $\nabla J_{i,H}(\theta) = \mathbb{E}_{\tau_i}[g_{i,H}(\tau_i, \theta)]$. In the heterogeneous setting, Proposition 2.3–(1), (2), and (3) still hold. However, due to heterogeneity, the last property concerning the unbiasedness and the variance of stochastic gradients holds only locally, for $g_{i,H}$ instead of $g_i$.

**Proposition 2.4.** *For all $i \in [n]$, let $\pi_{i,\theta}$ satisfy Assumptions 2.1 and 2.2. Then, Proposition 2.3-(1),(2),(3) are satisfied in the heterogeneous setting (4) and (5) for the functions $J$ and $J_H$ (follows from triangle's inequality).*
*4.     For $g_{i,H}$ defined in (5), we have $\nabla J_{i,H}(\theta) = \mathbb{E}_\tau[g_{i,H}(\tau, \theta)]$ and $\mathbb{E}\left[ \|g_{i,H}(\tau, \theta) - \nabla J_{i,H}(\theta)\|^2 \right] \leq \sigma^2$ for all $\theta \in \mathbb{R}^d, i \in [n]$.*

## 2.3 COMPUTATION AND COMMUNICATION TIMES

To illustrate our improvements over the previous state-of-the-art results, we make the following assumption. Notice that it is not required for the convergence of our methods.

> **Assumption 2.5.**
>
> - Computing a single stochastic policy gradient $g_H$ (or $g_{i,H}$) on agent $i$ requires at most
>
> $$h_i \; := \; \dot{h}_i \times H$$
>
>   seconds, where $\dot{h}_i$ denotes the time required to obtain the next state, and $H$ is the trajectory length. Without loss of generality, we assume that $\dot{h}_1 \leq \cdots \leq \dot{h}_n$, and consequently $h_1 \leq \cdots \leq h_n$.
> - In the centralized setup (with a server), transmitting a vector from an agent to and from the server takes at most $\kappa$ seconds. In the decentralized setup, transmitting vectors between agents (e.g., via AllReduce) also takes at most $\kappa$ seconds.

In order to compute a stochastic gradient $g_H$, an agent must generate a trajectory and collect the corresponding rewards. Since trajectories are generated sequentially in a Markov decision process, the time required grows linearly with the horizon length $H$. Therefore, it is natural to assume that the computation time is bounded by $\dot{h}_i \times H$, where $\dot{h}_i$ denotes the maximal time needed by agent $i$ to simulate or observe a single state transition.

The second condition is natural in distributed training and optimization, where communication between agents requires a non-negligible time $\kappa > 0$, for instance when performed over the Internet or via MPI-based message passing.

We consider heterogeneous computations $h_i$ and the communication time $\kappa$ to compare methods and highlight our contributions in the asynchronous setup. *All our new methods work without assuming these.* Moreover, Theorem 4.3 illustrates how our time complexity results can be generalized to arbitrary computation patterns.

## 3 New Methods: Rennala NIGT and Malenia NIGT

In this section, we present our new algorithm, Rennala NIGT (Algorithm 1 and Algorithm 2). Later, we present its heterogeneous version, Malenia NIGT (Algorithm 1 and Algorithm 3). They are inspired by (Cutkosky & Mehta, 2020; Fatkhullin et al., 2023; Tyurin & Richtárik, 2023). The core steps (Algorithm 1) are almost the same as in (Cutkosky & Mehta, 2020). Using the extrapolation steps, Cutkosky & Mehta (2020) showed that it is possible to improve the oracle complexity of the vanilla SGD method (Lan, 2020) under the additional assumption that the Hessian is smooth. It turns out that the oracle complexity from (Cutkosky & Mehta, 2020) can be improved with the STORM/MVR method (Cutkosky & Orabona, 2019), but the latter requires calculating stochastic gradients at two different points using the same random variable, which cannot easily be applied in the RL context due to the non-stationarity of the distribution.

To adapt Algorithm 1 to parallel and asynchronous scenarios, we follow the idea from (Tyurin & Richtárik, 2023; Tyurin et al., 2024a) and design the AggregateRennala procedure. The method broadcasts $\theta$ to all agents. Each agent $i$ then starts sampling $\tau_{i,1}$, obtaining the reward, and computing the stochastic gradient $g_H(\tau_{i,1}, \theta)$ locally. Next, the algorithm enters a loop and waits for any agent to complete these steps. Once an agent finishes, the algorithm increases the counter $i$ and instructs that agent to sample another trajectory and repeat the process. This continues until the *total* number of calculated stochastic gradients reaches $M$. Notice that, unlike (Lan et al., 2025), the agents can aggregate the stochastic gradients locally, thereby reducing the communication overhead. Finally, performing only one communication, the algorithm aggregates all vectors to $\bar{g}$ (e.g., via AllReduce), which Algorithm 1 uses to make the steps.

Our strategy offers several advantages: i) Rennala NIGT (Algorithm 1 and Algorithm 2) can be applied in both centralized and decentralized settings; ii) it is asynchronous-friendly, as Algorithm 2 is resilient to stragglers: if an agent is slow or even disconnected, the procedure is not delayed, since AggregateMalenia only needs to collect $M$ stochastic gradients from *all agents*. This will be formalized in Section 4. iii) it is also communication-efficient, as vector communication occurs only once at the end of Algorithm 2 (see Section 4.1). iv) finally, our theoretical guarantees are provably better than those of the previous results.

---

**Algorithm 1** Rennala NIGT or Malenia NIGT

1: **Input:** momentum $\eta$ and step size $\alpha$, starting point $\theta_0$, parameters $M_{\text{init}}$ and $M$, horizon $H$
2: Initialize $d_0 = \text{AggregateRennala}(\theta_0, M_{\text{init}}, H)$ $\qquad$ (or $= \text{AggregateMalenia}(\theta_0, M_{\text{init}}, H)$)
3: $\theta_1 = \theta_0 + \alpha \frac{d_0}{\|d_0\|}$
4: **for** $t = 1, 2, \ldots$ **do**
5: $\quad \widetilde{\theta}_t = \theta_t + \frac{1-\eta}{\eta}(\theta_t - \theta_{t-1})$
6: $\quad g_t = \text{AggregateRennala}(\widetilde{\theta}_t, M, H)$ $\qquad$ (or $= \text{AggregateMalenia}(\widetilde{\theta}_t, M, H)$)
7: $\quad d_t = (1-\eta)d_{t-1} + \eta g_t$
8: $\quad \theta_{t+1} = \theta_t + \alpha \frac{d_t}{\|d_t\|}$
9: **end for**

---

| **Algorithm 2** AggregateRennala($\theta, M, H$) | **Algorithm 3** AggregateMalenia($\theta, M, H$) |
|---|---|
| 1: Init $\bar{g} = 0 \in \mathbb{R}^d$ and $i = 1$ | 1: Init $\bar{g}_i = 0 \in \mathbb{R}^d$ and $M_i = 0$ for all $i \in [n]$ |
| 2: Broadcast $\theta$ to all agents | 2: Broadcast $\theta$ to all agents |
| 3: Each agent $i$ starts sampling $\tau_{i,1} \sim p(\cdot\|\pi_\theta)$ and calculating $g_H(\tau_{i,1}, \theta)$ | 3: Each agent $i$ starts sampling $\tau_{i,1} \sim p_i(\cdot\|\pi_{i,\theta})$ and calculating $g_{i,H}(\tau_{i,1}, \theta)$ |
| 4: **while** $i \leq M$ **do** | 4: **while** $(1/n \sum_{i=1}^n 1/M_i)^{-1} < M/n$ **do** |
| 5: $\quad$ Wait for $g_H(\tau_{j,k}, \theta)$ from an agent $j$ | 5: $\quad$ Wait for $g_{j,H}(\tau_{j,k}, \theta)$ from an agent $j$ |
| 6: $\quad \bar{g} = \bar{g} + \frac{1}{M}g_H(\tau_{j,k}, \theta); i = i+1$ | 6: $\quad \bar{g}_j = \bar{g}_j + g_{j,H}(\tau_{j,k}, \theta); M_j = M_j + 1$ |
| 7: $\quad$ Agent $j$ starts sampling $\tau_{j,k+1} \sim p(\cdot\|\pi_\theta)$ and calculating $g_H(\tau_{j,k+1}, \theta)$ | 7: $\quad$ Agent $j$ starts sampling $\tau_{j,k+1} \sim p_j(\cdot\|\pi_{j,\theta})$ and calculating $g_{j,H}(\tau_{j,k+1}, \theta)$ |
| 8: **end while** | 8: **end while** |
| 9: Stop all calculations | 9: Stop all calculations |
| 10: Return $\bar{g}$ $\qquad$ (e.g., via AllReduce) | 10: Return $1/n \sum_{i=1}^n \bar{g}_i/M_i$ $\quad$ (e.g., via AllReduce) |

**Note:** The agents can locally aggregate $g_H$ and $g_{i,H}$, after which the algorithm can perform a single AllReduce call to collect all computed gradients. The trajectories $\{\tau_{i,j}\}$ are statistically independent.

## 4 TIME COMPLEXITY IN THE HOMOGENEOUS SETUP

**Theorem 4.1.** *Let Assumptions 2.1 and 2.2 hold. Consider* Rennala NIGT *(Algorithm 1 and Algorithm 2) in the homogeneous setup, or* Malenia NIGT *(Algorithms 1 and Algorithm 3) in the heterogeneous setup. Let* $\eta = \min\left\{\frac{M\varepsilon^2}{64\sigma^2}, \frac{1}{2}\right\}$, $\alpha = \min\left\{\frac{\varepsilon}{8L_g}, \frac{\eta\sqrt{\varepsilon}}{4\sqrt{L_h}}\right\}$, $H = \max\left\{\log_\gamma\left(\frac{\varepsilon\eta}{64\max\{D_g, \alpha D_h\}}\right), 1\right\} = \tilde{\mathcal{O}}(1/(1-\gamma))$. *Let* $\bar{\theta}_T$ *be a uniformly sampled iterate from* $\{\theta_0, \cdots, \theta_{T-1}\}$. *Then* $\mathbb{E}\|\nabla J(\bar{\theta}_T)\| \leq \varepsilon$ *after* $T = \mathcal{O}\left(\frac{L_g\Delta}{\varepsilon^2} + \frac{\sqrt{L_h}\Delta}{\varepsilon^{3/2}} + \frac{\sigma}{\sqrt{M_{init}}\varepsilon} + \frac{\sigma^3}{M\sqrt{M_{init}}\varepsilon^3} + \frac{\sigma^2\sqrt{L_h}\Delta}{M\varepsilon^{7/2}}\right)$ *global iterations.*

**Notice that Theorem 4.1 does not rely on Assumption 2.5; convergence is guaranteed even without Assumption 2.5.** It is an auxiliary result that holds for any choice of $M$ and $M_{\text{init}}$. We now establish the time complexity of Rennala NIGT in Theorem 4.1, our first main result.

**Theorem 4.2.** *Consider the results and assumptions of Theorem 4.1. Additionally, consider that Assumption 2.5 holds with* $\kappa = 0$ *(i.e., communication is free). Taking* $M_{init} = \max\left\{\lceil\frac{\sigma^2}{\varepsilon^2}\rceil, 1\right\}$ *and* $M = \max\left\{\lceil(\frac{\sigma^2}{\varepsilon^2} + \frac{\sigma^2\sqrt{L_h}\Delta}{\varepsilon^{7/2}})/(\frac{L_g\Delta}{\varepsilon^2} + \frac{\sqrt{L_h}\Delta}{\varepsilon^{3/2}})\rceil, 1\right\}$, *the time required to find an* $\varepsilon-$ *stationary point by* Rennala NIGT *(Algorithms 1 and 2) is*

$$\tilde{\mathcal{O}}\left(\frac{1}{1-\gamma}\min_{m\in[n]}\left[\left(\frac{1}{m}\sum_{i=1}^m \frac{1}{h_i}\right)^{-1}\left(\frac{L_g\Delta}{\varepsilon^2} + \frac{\sqrt{L_h}\Delta}{\varepsilon^{3/2}} + \frac{\sigma^2}{m\varepsilon^2} + \frac{\sigma^2\sqrt{L_h}\Delta}{m\varepsilon^{7/2}}\right)\right]\right). \qquad (6)$$

We now compare this result with (Lan et al., 2025). Although that paper requires the impractical assumption that generated trajectories have infinite horizons, we assume that they also require $\dot{h}_i \times H$ seconds to generate one trajectory for agent $i$. Then, they obtain at least the time complexity

$\tilde{\mathcal{O}}\big( \left(1/n \sum_{i=1}^{n} 1/\dot{h}_i\right)^{-1} \left(n^{4/3}/\varepsilon^{7/3} + 1/n\varepsilon^{7/2}\right) \big)$, up to $\varepsilon$, $n$, and $\dot{h}_i$ constant factors. If $n$ is large, which is the case in federated learning and distributed optimization, their complexity can be arbitrarily large due to the $n^{4/3}$ and $\left(1/n \sum_{i=1}^{n} 1/\dot{h}_i\right)^{-1}$ dependencies[3]. Notice that our complexity (6) can only decrease with larger $n$ due to the $\tilde{\mathcal{O}}\big( \min_{m \in [n]} \big[ \left(\frac{1}{m} \sum_{i=1}^{m} 1/\dot{h}_i\right)^{-1} \left(1/\varepsilon^2 + 1/m\varepsilon^{7/2}\right) \big] \big)$ dependency. In Section 4.1, we show that the gap is even larger when we start taking into account the communication factor.

Our time complexity (6) has a harmonic-like dependency $\tilde{\mathcal{O}}\big( \min_{m \in [n]} \big[ \left(1/m \sum_{i=1}^{m} 1/\dot{h}_i\right)^{-1} \left(A + B/m\right) \big] \big)$ on $\{\dot{h}_i\}$ for $A := {}^{L_g\Delta}/\varepsilon^2 + \sqrt{L_h}\Delta/\varepsilon^{3/2}$ and $B := \sigma^2/\varepsilon^2 + \sigma^2\sqrt{L_h}\Delta/\varepsilon^{7/2}$. To the best of our knowledge, this is the current state-of-the-art computational complexity for maximizing (1). It has many nice properties: i) it is robust to stragglers. If we take $\dot{h}_n \to \infty$, this complexity starts ignoring the slowest agent and becomes $\tilde{\mathcal{O}}\big( \min_{m \in [n-1]} \big[ \left(1/m \sum_{i=1}^{m} 1/\dot{h}_i\right)^{-1} \left(A + B/m\right) \big] \big)$; ii) since the harmonic mean is less than or equal to the maximum term, this complexity is much better than the time complexity of the naive synchronized distributed version of NIGT with the complexity $\tilde{\mathcal{O}}\big( \max_{i \in [n]} \dot{h}_i \left(A + B/n\right) \big)$, where all agents synchronize after each has computed one stochastic gradient. Notice that we can easily generalize our result when the times are non-static.

**Theorem 4.3.** *Consider the results and assumptions of Theorem 4.1. Additionally, consider that computing a single stochastic policy gradient $g_H$ on agent $i$ in iteration $t$ requires at most $\dot{h}_{t,i} \times H$ seconds. Taking $M_{init} = \max\left\{\left\lceil \frac{\sigma^2}{\varepsilon^2} \right\rceil, 1\right\}$ and $M = \max\left\{\left\lceil \left(\frac{\sigma^2}{\varepsilon^2} + \frac{\sigma^2\sqrt{L_h}\Delta}{\varepsilon^{7/2}}\right)/\left(\frac{L_g\Delta}{\varepsilon^2} + \frac{\sqrt{L_h}\Delta}{\varepsilon^{3/2}}\right) \right\rceil, 1\right\}$, the time required to find an $\varepsilon$–stationary point by* Rennala NIGT *(Algorithms 1 and 2) is $\tilde{\mathcal{O}}\big(1/1-\gamma \sum_{t=1}^{T} \min_{m \in [n]} \big[ \left(\frac{1}{m} \sum_{i=1}^{m} 1/\dot{h}_{t,\pi_{t,i}}\right)^{-1} \left(M/m + 1\right) \big] + 1/1-\gamma \min_{m \in [n]} \big[ \left(\frac{1}{m} \sum_{i=1}^{m} 1/\dot{h}_{t,\pi_{0,i}}\right)^{-1} \left(M_{init}/m + 1\right) \big]\big)$, where $T = \mathcal{O}\left(L_g\Delta/\varepsilon^2 + \sqrt{L_h}\Delta/\varepsilon^{3/2}\right)$ and $\pi_{t,i}$ is a permutation such that $\dot{h}_{t,\pi_{t,1}} \leq \cdots \leq \dot{h}_{t,\pi_{t,n}}$.*

To simplify the discussion, we further focus on Assumption 2.5 and assume that the bounds on computation times are $\{h_i\}$.

## 4.1 TIME COMPLEXITY WITH COMMUNICATION TIMES

We now generalize Theorem 4.2 by taking into account the communication time $\kappa$:

**Theorem 4.4.** *Consider the results and assumptions of Theorem 4.1. Additionally, consider that Assumption 2.5 holds. Taking $M_{init} = \max\left\{\left\lceil \frac{\sigma^2}{\varepsilon^2} \right\rceil, 1\right\}$ and $M = \max\left\{\left\lceil \left(\frac{\sigma^2}{\varepsilon^2} + \frac{\sigma^2\sqrt{L_h}\Delta}{\varepsilon^{7/2}}\right)/\left(\frac{L_g\Delta}{\varepsilon^2} + \frac{\sqrt{L_h}\Delta}{\varepsilon^{3/2}}\right) \right\rceil, 1\right\}$, the time required to find an $\varepsilon$–stationary point by* Rennala NIGT *(Algorithms 1 and 2) is*

$$\tilde{\mathcal{O}}\left( \kappa\left(\frac{L_g\Delta}{\varepsilon^2} + \frac{\sqrt{L_h}\Delta}{\varepsilon^{3/2}}\right) + \frac{1}{1-\gamma} \min_{m \in [n]} \left[ \left(\frac{1}{m} \sum_{i=1}^{m} \frac{1}{h_i}\right)^{-1} \left(\frac{L_g\Delta}{\varepsilon^2} + \frac{\sqrt{L_h}\Delta}{\varepsilon^{3/2}} + \frac{\sigma^2}{m\varepsilon^2} + \frac{\sigma^2\sqrt{L_h}\Delta}{m\varepsilon^{7/2}}\right) \right] \right).$$

Compared to (6), we obtain an additional time term $\kappa\left(L_g\Delta/\varepsilon^2 + \sqrt{L_h}\Delta/\varepsilon^{3/2}\right)$, which accounts for the communication time complexity. Recall that the method of Lan et al. (2025) performs asynchronous updates, where each agents independently send gradients to the server, which is non-communication-efficient. For small $\varepsilon$, in Lan et al. (2025), the fastest agent sends at least $\mathcal{O}(\max\{n^{4/3}\varepsilon^{-7/3}, n^{-1}\varepsilon^{-7/2}\})$ stochastic gradients (according to their Theorem 5.2). Consequently, their communication time complexity is at least[4] $\mathcal{O}(\kappa\varepsilon^{-3})$, whereas our algorithm achieves a significantly better dependence on $\varepsilon$, namely $\mathcal{O}(\kappa\varepsilon^{-2})$. In the extreme case, when only the fastest agent participates and contributes, AFedPG reduces to a non-distributed stochastic method in which a single agent sends a gradient for every oracle call. In this setting, AFedPG provably requires at least $\Omega(\varepsilon^{-7/2})$ communications (Arjevani et al., 2020), resulting in an even larger gap compared to our complexity of $\mathcal{O}(\varepsilon^{-2})$. Moreover, our method supports AllReduce, an important feature in practical engineering scenarios.

---

[3] For instance, take $\dot{h}_i = h$, then it reduces to $\dot{h}\left(n^{4/3}/\varepsilon^{7/3} + 1/n\varepsilon^{7/2}\right) \overset{n \to \infty}{\to} \infty$. As a concrete example, if $\varepsilon = 0.0001$, the first term $n^{4/3}/\varepsilon^{7/3}$ already dominates when $n = 100$. In practice, the number of computational resources $n$ continues to grow toward 10K–100K, causing the complexity to increase with $n$ as well.

[4] since $\max\{n^{4/3}\varepsilon^{-7/3}, n^{-1}\varepsilon^{-7/2}\} = \Omega(\varepsilon^{-3})$

In Section G, we prove a lower bound for functions and stochastic gradients satisfying Proposition 2.3. Although we do not establish a lower bound under Assumptions 2.1 and 2.2, we believe our result still reflects the fundamental lower bound of the considered task, since all recent state-of-the-art results (Fatkhullin et al., 2023; Lan et al., 2025), including ours, rely solely on Proposition 2.3. Comparing Theorems 4.4 and G.1, one can see that there is still a gap, for instance, between $\kappa \varepsilon^{-2}$ and $\kappa \varepsilon^{-12/7}$. We obtain the latter term by reducing our problem to the lower bound of Carmon et al. (2021) for $(L_g, L_h)$–twice smooth functions. To the best of our knowledge, it remains an open problem whether the $\varepsilon^{-12/7}$ rate can be achieved, even in the non-distributed deterministic case.

## 5 TIME COMPLEXITIES IN THE HETEROGENEOUS SETUP

In the heterogeneous setup, we consider Algorithm 3 instead of Algorithm 2. Algorithm 3 is also an asynchronous aggregation scheme of stochastic gradients adapted to the heterogeneous setting. Unlike Algorithm 2, which is specialized for the homogeneous setup, Algorithm 3 ensures that the returned vector is unbiased in the heterogeneous setup: $\mathbb{E}\left[1/n \sum_{i=1}^{n} \bar{g}_i / M_i\right] = 1/n \sum_{i=1}^{n} J_{i,H}(\theta) = J_H(\theta)$. For Malenia NIGT, we can prove the following result:

> **Theorem 5.1.** *Consider the results and assumptions of Theorem 4.1. Additionally, consider that Assumption 2.5 holds. Taking $M_{init} = \max\left\{\lceil \frac{\sigma^2}{\varepsilon^2}\rceil, 1\right\}$ and $M = \max\left\{\lceil\left(\frac{\sigma^2}{\varepsilon^2} + \frac{\sigma^2\sqrt{L_h}\Delta}{\varepsilon^{7/2}}\right)/\left(\frac{L_g\Delta}{\varepsilon^2} + \frac{\sqrt{L_h}\Delta}{\varepsilon^{3/2}}\right)\rceil, 1\right\}$, the time required to find an $\varepsilon$–stationary point by* Malenia NIGT *(Algorithms 1 and 3) is*
> $$\tilde{\mathcal{O}}\left(\kappa\left(\frac{L_g\Delta}{\varepsilon^2} + \frac{\sqrt{L_h}\Delta}{\varepsilon^{3/2}}\right) + \frac{1}{1-\gamma}\left[\dot{h}_n\left(\frac{L_g\Delta}{\varepsilon^2} + \frac{\sqrt{L_h}\Delta}{\varepsilon^{3/2}}\right) + \left(\frac{1}{n}\sum_{i=1}^{n}\dot{h}_i\right)\left(\frac{\sigma^2}{n\varepsilon^2} + \frac{\sigma^2\sqrt{L_h}\Delta}{n\varepsilon^{7/2}}\right)\right]\right).$$

To the best of our knowledge, this result represents the current state-of-the-art complexity for RL problems in the heterogeneous, distributed, and asynchronous setting. Compared to Theorem 4.2, which has a harmonic-like dependency, the dependence on $\{\dot{h}_i\}$ in Theorem 5.1 is mean-like in the small-$\varepsilon$ regime. This behavior is expected, as the heterogeneous case is more challenging due to agents operating with different distributions and environments. However, the term related to the communication time complexity is the same.

## 6 SUMMARY OF EXPERIMENTS

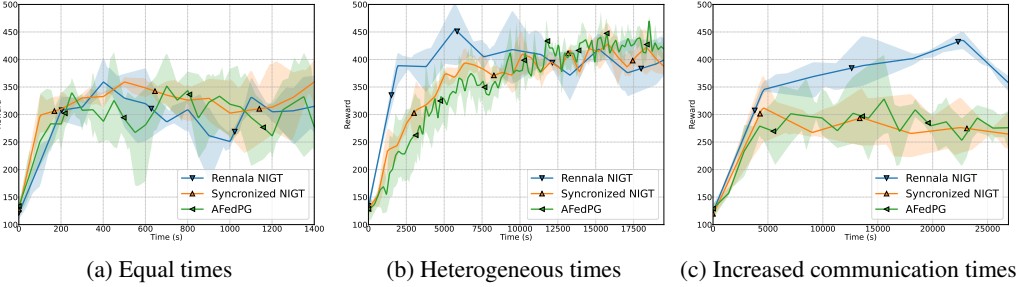

(a) Equal times  (b) Heterogeneous times  (c) Increased communication times

Figure 1: Experiments on Humanoid-v4 with increasing heterogeneity of times (from left to right).

In this section, we empirically test the performance of Rennala NIGT and compare it with the previous state-of-the-art method by (Lan et al., 2025) and the synchronized version of NIGT (Synchronized NIGT). Following previous works (Lan et al., 2025; Fatkhullin et al., 2023), we focus on the MuJoCo tasks (Todorov et al., 2012). We defer details, parameters, and additional experiments to Section H. Figure 1 shows the main results: i) when the computation times are equal, all methods exhibit almost the same performance, which is expected since they reduce to nearly the same algorithm; ii) when the computation times are heterogeneous, Rennala NIGT converges faster than other methods; iii) when the computation times are heterogeneous and the communication times are large, Rennala NIGT is the only robust method, and the gap increases even more. These experiments, together with the additional experiments from Section H, support our theoretical results.

## 7 Universal Computation Model

Following (Tyurin, 2025; Maranjyan et al., 2025), for completeness, we can extend the previously derived time complexities to arbitrary computational dynamics.

---

**Assumption 7.1.**

- We define a non-negative and continuous almost everywhere *computation power* function $v_i : \mathbb{R}_+ \to \mathbb{R}_+$ for all agent $i \in [n]$. The number of stochastic policy gradient that agent $i$ can compute between times $t_0$ and $t_1$ is

$$N_i(t_0, t_1) := \left\lfloor \int_{t_0}^{t_1} v_i(\tau) d\tau \right\rfloor. \tag{7}$$

- For simplicity, we ignore the communication time.

---

Using these computational powers, we can formalize the changing computation behaviors of agents, taking into account random fluctuations, different trends, and disconnections in a more flexible way. In particular, when $v_i(\tau) = v_i \in \mathbb{R}_+$ is constant, $N_i(t_0, t_1) = \lfloor v_i \times (t_1 - t_0) \rfloor$, and if agent $i$ starts calculating at time $t_0$, then it will compute one policy gradient after $t_0 + 1/v_i$ seconds because $N_i(t_0, t_0 + 1/v_i) = 1$, two after $t_0 + 2/v_i$ seconds, and so forth. This example reduces to Assumption 2.5 with $h_i \equiv 1/v_i$. However, Assumption 7.1 allows us to capture virtually any computational scenarios (see examples in (Tyurin, 2025)).

---

**Theorem 7.2.** *Consider the results and assumptions of Theorem 4.1. Additionally, consider that Assumption 7.1 holds. Taking $M_{init} = \max\left\{\left\lceil\frac{\sigma^2}{\varepsilon^2}\right\rceil, 1\right\}$ and $M = \max\left\{\left\lceil\left(\frac{\sigma^2}{\varepsilon^2} + \frac{\sigma^2\sqrt{L_h}\Delta}{\varepsilon^{7/2}}\right)/\left(\frac{L_g\Delta}{\varepsilon^2} + \frac{\sqrt{L_h}\Delta}{\varepsilon^{3/2}}\right)\right\rceil, 1\right\}$, the time required to find an $\varepsilon$-stationary point by* Rennala NIGT *(Algorithms 1 and 2) is $t_T$ seconds, where $T = \mathcal{O}\left(\frac{L_g\Delta}{\varepsilon^2} + \frac{\sqrt{L_h}\Delta}{\varepsilon^{3/2}}\right)$ and the sequence $t_k$ is defined recursively:*

$$t_k = \min\left\{t : \sum_{i=1}^{n} N_i(t_{k-1}, t) \geq M\right\} \qquad \forall k \geq 1 \tag{8}$$

*with $t_0 = \min\left\{t : \sum_{i=1}^{n} N_i(0, t) \geq M_{init}\right\}$.*

---

One can show that this theorem admits an analytical formula and reduces to Theorem 4.2 when $\{v_i\}$ are constant functions. In general, to find $t_T$, one should solve (8) for each particular choice of $v_i$, and it is always possible to do so numerically. First, find the smallest $t \geq 0$ such that $\sum_{i=1}^{n} N_i(0, t) \geq M_{init}$, where $\sum_{i=1}^{n} N_i(0, t)$ is the number of gradients that the agents can compute in parallel after $t$ seconds. Then, repeat the same procedure recursively for (8). We can also extend Theorem 5.1 (ignoring communication times):

---

**Theorem 7.3.** *Consider the results and assumptions of Theorem 4.1. Additionally, consider that Assumption 7.1 holds. Taking $M_{init} = \max\left\{\left\lceil\frac{\sigma^2}{\varepsilon^2}\right\rceil, 1\right\}$ and $M = \max\left\{\left\lceil\left(\frac{\sigma^2}{\varepsilon^2} + \frac{\sigma^2\sqrt{L_h}\Delta}{\varepsilon^{7/2}}\right)/\left(\frac{L_g\Delta}{\varepsilon^2} + \frac{\sqrt{L_h}\Delta}{\varepsilon^{3/2}}\right)\right\rceil, 1\right\}$, the time required to find an $\varepsilon$-stationary point by* Malenia NIGT *(Algorithms 1 and 3) is $t_T$ seconds, where $T = \mathcal{O}\left(\frac{L_g\Delta}{\varepsilon^2} + \frac{\sqrt{L_h}\Delta}{\varepsilon^{3/2}}\right)$ and the sequence $t_k$ is defined recursively:*

$$t_k = \min\left\{t : \left(\frac{1}{n}\sum_{i=1}^{n}\frac{1}{N_i(t_{k-1}, t)}\right)^{-1} \geq \frac{M}{n}\right\} \qquad \forall k \geq 1 \tag{9}$$

*with $t_0 = \min\left\{t : \left(\frac{1}{n}\sum_{i=1}^{n}\frac{1}{N_i(0, t)}\right)^{-1} \geq \frac{M_{init}}{n}\right\}$.*

---

### Acknowledgements

The work was supported by the grant for research centers in the field of AI provided by the Ministry of Economic Development of the Russian Federation in accordance with the agreement 000000C313925P4F0002 and the agreement №139-10-2025-033.

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

## A  GLOBAL CONVERGENCE

Under additional technical Assumptions 2.1 and 4.6 from (Ding et al., 2022) about the Fisher information matrix induced by the policy $\pi_\theta$ and the initial state distribution, following previous works (Ding et al., 2022; Fatkhullin et al., 2023; Lan et al., 2025), we can guarantee global convergence up to an $\sqrt{2}\varepsilon'/\sqrt{\mu}$ neighborhood:

**Theorem A.1.** *Let Assumptions 2.1, 2.2, and Assumptions 2.1 and 4.6 from (Ding et al., 2022) hold. Consider* Rennala NIGT *(Algorithm 1 and Algorithm 2) in the homogeneous setup, or* Malenia NIGT *(Algorithms 1 and Algorithm 3) in the heterogeneous setup. Let $M_{init} = \max\left\{\left\lceil\frac{\sigma^2}{\mu\varepsilon^2}\right\rceil, 1\right\}$, $M = \max\left\{\left\lceil\left(\frac{\sigma^2}{\mu\varepsilon^2} + \frac{\sigma^2\sqrt{L_h}}{\mu^{7/4}\varepsilon^{5/2}}\right)/\left(\frac{L_g}{\mu\varepsilon} + \frac{\sqrt{L_h}}{\mu^{3/4}\sqrt{\varepsilon}}\right)\right\rceil, 1\right\}$, $\eta = \min\left\{\frac{M\mu\varepsilon^2}{64\sigma^2}, \frac{1}{2}\right\}$, $\alpha = \min\left\{\frac{\sqrt{\mu\varepsilon}}{8L_g}, \frac{\eta\mu^{1/4}\sqrt{\varepsilon}}{4\sqrt{L_h}}, \frac{1}{\sqrt{2\mu}}, \frac{\varepsilon\eta\sqrt{M_{init}}}{8\sigma}\right\}$, $H = \max\left\{\log_\gamma\left(\frac{\sqrt{\mu}\varepsilon\eta}{64\max\{D_g, \alpha D_h\}}\right), 1\right\}$. Then $\mathbb{E}\left[J^* - J(\theta_T)\right] \leq \varepsilon + \frac{\sqrt{2}\varepsilon'}{\sqrt{\mu}}$ after $T = \mathcal{O}\left(\left(\frac{L_g}{\mu\varepsilon} + \frac{\sqrt{L_h}}{\mu^{3/4}\sqrt{\varepsilon}}\right)\log\left(\frac{\Delta}{\varepsilon}\right)\right)$ global iterations, where $\varepsilon' := \frac{\mu_F\sqrt{\varepsilon_{bias}}}{M_g(1-\gamma)}$ and $\mu := \frac{\mu_F^2}{2M_g^2}$. Parameter $\varepsilon_{bias}$ is an approximation error and $\mu_F$ is the smallest eigenvalue of the Fisher information matrix induced by the policy $\pi_\theta$ and and the initial state distribution (Ding et al., 2022).*

In this case, the time complexities of Rennala NIGT and Malenia NIGT follow the same structure as in Sections 4.1 and 5:

**Theorem A.2.** *Consider the results and assumptions of Theorem A.1. Additionally, consider that Assumption 2.5 holds. The time required to find an $\varepsilon$–solution up to $\sqrt{2}\varepsilon'/\sqrt{\mu}$ neighborhood by* Rennala NIGT *(Algorithms 1 and 2) in the homogeneous setup is*

$$\tilde{\mathcal{O}}\left(\kappa\left(\frac{L_g}{\mu\varepsilon} + \frac{\sqrt{L_h}}{\mu^{3/4}\sqrt{\varepsilon}}\right) + \frac{1}{1-\gamma}\min_{m\in[n]}\left[\left(\frac{1}{m}\sum_{i=1}^m \frac{1}{h_i}\right)^{-1}\left(\frac{L_g}{\mu\varepsilon} + \frac{\sqrt{L_h}}{\mu^{3/4}\sqrt{\varepsilon}} + \frac{\sigma^2}{m\mu\varepsilon^2} + \frac{\sigma^2\sqrt{L_h}}{m\mu^{7/4}\varepsilon^{5/2}}\right)\right]\right).$$

**Theorem A.3.** *Consider the results and assumptions of Theorem A.1. Additionally, consider that Assumption 2.5 holds. The time required to find an $\varepsilon$–solution up to $\sqrt{2}\varepsilon'/\sqrt{\mu}$ neighborhood by* Malenia NIGT *(Algorithms 1 and 3) in the heterogeneous setup is*

$$\tilde{\mathcal{O}}\left(\kappa\left(\frac{L_g}{\mu\varepsilon} + \frac{\sqrt{L_h}}{\mu^{3/4}\sqrt{\varepsilon}}\right) + \frac{1}{1-\gamma}\left[\dot{h}_n\left(\frac{L_g}{\mu\varepsilon} + \frac{\sqrt{L_h}}{\mu^{3/4}\sqrt{\varepsilon}}\right) + \left(\frac{1}{n}\sum_{i=1}^n \dot{h}_i\right)\left(\frac{\sigma^2}{n\mu\varepsilon^2} + \frac{\sigma^2\sqrt{L_h}}{n\mu^{7/4}\varepsilon^{5/2}}\right)\right]\right).$$

## B  NOTATIONS

$\mathbb{N} := \{1, 2, \dots\}$; $[n] := \{1, \dots n\}$; $\|\cdot\|$ is the Euclidean norm; $\langle x, y \rangle = \sum_{i=1}^d x_i y_i$ is the standard dot product; $\|A\|$ is the standard spectral norm for all $A \in \mathbb{R}^{d\times d}$; $g = \mathcal{O}(f)$ : there exists $C > 0$ such that $g(z) \leq C \times f(z)$ for all $z \in \mathcal{Z}$; $g = \Omega(f)$ : there exists $C > 0$ such that $g(z) \geq C \times f(z)$ for all $z \in \mathcal{Z}$; $g = \Theta(f)$ : if $g = \mathcal{O}(f)$ and $g = \Omega(f)$ : $g \simeq h$ : $g$ and $h$ are equal up to a universal positive constant; $\tilde{\mathcal{O}}, \tilde{\Omega}, \tilde{\Theta}$ the same as $\mathcal{O}, \Omega, \Theta$, but up to logarithmic factors.

## C  USEFUL LEMMAS

The following lemmas provide the time complexities of collecting stochastic gradients in Algorithms 2 and 3.

**Lemma C.1.** *Under Assumption 2.5, Algorithms 2 returns the output vector after at most*

$$\mathcal{O}\left(\kappa + \min_{m\in[n]}\left[\left(\sum_{i=1}^m \frac{1}{h_i}\right)^{-1}(M+m)\right]\right) = \mathcal{O}\left(\kappa + \min_{m\in[n]}\left[\left(\frac{1}{m}\sum_{i=1}^m \frac{1}{h_i}\right)^{-1}\left(\frac{M}{m}+1\right)\right]\right)$$

*seconds.*

*Proof.* Let

$$t = \min_{m \in [n]} \left( \left( \sum_{i=1}^{m} \frac{1}{h_i} \right)^{-1} (M + m) \right).$$

As soon as some agent finishes computing a stochastic policy gradient, it immediately starts computing the next one. Hence, by time $t$, the agents together will have processed at least

$$\sum_{i=1}^{n} \left\lfloor \frac{t}{h_i} \right\rfloor$$

stochastic gradients. Let

$$m^* = \arg \min_{m \in [n]} \left( \left( \sum_{i=1}^{m} \frac{1}{h_i} \right)^{-1} (M + m) \right).$$

Using $\lfloor x \rfloor \geq x - 1$ for all $x \geq 0$, we obtain

$$\sum_{i=1}^{n} \left\lfloor \frac{t}{h_i} \right\rfloor \geq \sum_{i=1}^{m^*} \left\lfloor \frac{t}{h_i} \right\rfloor \geq \sum_{i=1}^{m^*} \frac{t}{h_i} - m^* = \left( \sum_{i=1}^{m^*} \frac{1}{h_i} \right) \left( \left( \sum_{i=1}^{m^*} \frac{1}{h_i} \right)^{-1} (M + m^*) \right) - m^* = M.$$

Therefore, by time $t$ the algorithm has collected at least $M$ stochastic gradients and exits the inner loop. Returning the output vector then requires a single communication/aggregation, which by Assumption 2.5 takes at most $\kappa$ seconds (via a server or AllReduce for instance). In the centralized setting, broadcasting would also take at most $\kappa$ seconds. Thus, Algorithm 2 returns after at most $t + 2\kappa = \mathcal{O}\left( \kappa + \min_{m \in [n]} \left[ \left( \sum_{i=1}^{m} \frac{1}{h_i} \right)^{-1} (M + m) \right] \right)$ seconds. $\qquad\square$

**Lemma C.2.** *Under Assumption 2.5, Algorithms 3 returns the output vector after at most*

$$\mathcal{O}\left( \kappa + h_n + \left( \frac{1}{n} \sum_{i=1}^{n} h_i \right) \frac{M}{n} \right)$$

*seconds.*

*Proof.* By Assumption 2.5, broadcasting $\theta$ takes at most $\kappa$ seconds. After that, the agents compute in parallel and after $h_n$ seconds each agent has completed at least one gradient. Thus, $M_i \geq 1$ for all $i \in [n]$ after $h_n$ seconds. Let us take

$$t = \left( \frac{1}{n} \sum_{i=1}^{n} h_i \right) \frac{M}{n}.$$

For all $i \in [n]$, after $h_n + t$ seconds, agent $i$ produces $M_i(t)$ gradients, where $M_i(t)$ satisfy the inequality

$$M_i(t) \geq 1 + \left\lfloor \frac{t}{h_i} \right\rfloor.$$

since agent $i$ can calculate at least $\left\lfloor \frac{t}{h_i} \right\rfloor$ gradients after $t$ seconds. Using $\lfloor x \rfloor \geq x - 1$ for all $x \in \mathbb{R}$,

$$\frac{1}{M_i(t)} \leq \frac{1}{1 + \lfloor t/h_i \rfloor} \leq \frac{h_i}{t}.$$

Summing over $i$ yields

$$\sum_{i=1}^{n} \frac{1}{M_i(t)} \leq \frac{1}{t} \sum_{i=1}^{n} h_i = \frac{n^2}{M}$$

and

$$\left( \frac{1}{n} \sum_{i=1}^{n} \frac{1}{M_i(t)} \right)^{-1} \geq \frac{M}{n}$$

Thus, by time $\kappa + h_n + t$ the while-condition $\left(\frac{1}{n}\sum_{i=1}^{n}\frac{1}{M_i}\right)^{-1} < \frac{M}{n}$ becomes false and the loop terminates. Returning the output vector then requires at most an additional $\kappa$ seconds. In total, it takes

$$\mathcal{O}\left(\kappa + h_n + \left(\frac{1}{n}\sum_{i=1}^{n}h_i\right)\frac{M}{n}\right)$$

seconds. $\qquad\square$

## D  PROOF OF THEOREMS

We start with supporting lemmas and then apply them in the proof of the theorem.

**Lemma D.1.** *Let Assumption 2.1 and 2.2 hold, $\{d_t\}$ is any sequence of vectors from $\mathbb{R}^d$ and $\theta_{t+1} = \theta_t + \alpha\frac{d_t}{\|d_t\|}$, where $\theta_0 \in \mathbb{R}^d$ and we use the standard convention $0/\|0\| = 0$. Then*

$$-J(\theta_{t+1}) \le -J(\theta_t) - \alpha\|\nabla J(\theta_t)\| + 2\alpha\|d_t - \nabla J(\theta_t)\| + \frac{L_g\alpha^2}{2}. \tag{10}$$

*for every integer $t \ge 0$.*

*Proof.* Using Proposition 2.3-(1) (or Proposition 2.4-(1)) and the update rule, we get

$$-J(\theta_{t+1}) \le -J(\theta_t) - \langle\nabla J(\theta_t), \theta_{t+1} - \theta_t\rangle + \frac{L_g}{2}\|\theta_{t+1} - \theta_t\|^2$$

$$= -J(\theta_t) - \frac{\alpha}{\|d_t\|}\langle\nabla J(\theta_t), d_t\rangle + \frac{L_g}{2}\left(\frac{\alpha}{\|d_t\|}\right)^2\|d_t\|^2$$

$$= -J(\theta_t) - \frac{\alpha}{\|d_t\|}\|d_t\|^2 - \frac{\alpha}{\|d_t\|}\langle\nabla J(\theta_t) - d_t, d_t\rangle + \frac{L_g\alpha^2}{2}$$

$$\le -J(\theta_t) - \alpha\|d_t\| + \alpha\|d_t - \nabla J(\theta_t)\| + \frac{L_g\alpha^2}{2},$$

where we use the C-S inequality. Using $\|\nabla J(\theta_t)\| \le \|d_t\| + \|d_t - \nabla J(\theta_t)\|$, we have

$$-J(\theta_{t+1}) \le -J(\theta_t) - \alpha\|\nabla J(\theta_t)\| + 2\alpha\|d_t - \nabla J(\theta_t)\| + \frac{L_g\alpha^2}{2}.$$

$\qquad\square$

Let $\mathbb{E}_t[\cdot]$ be the conditional expectation condition on all randomness up to $t^{\text{th}}$ iteration. In the next lemma, we bound $\sum_{t=0}^{T-1}\mathbb{E}[\|d_t - \nabla J_H(\theta_t)\|]$.

**Lemma D.2.** *Let Assumptions 2.1 and 2.2 hold. Consider Algorithms 1 and assume that*

$$\mathbb{E}[d_0] = \nabla J_H(\theta_0) \text{ and } \mathbb{E}\left[\|d_0 - \nabla J_H(\theta_0)\|^2\right] \le \frac{\sigma^2}{M_{init}} \tag{11}$$

*and*

$$\mathbb{E}_t[g_t] = \nabla J_H(\tilde{\theta}_t) \text{ and } \mathbb{E}_t\left[\left\|g_t - \nabla J_H(\tilde{\theta}_t)\right\|^2\right] \le \frac{\sigma^2}{M} \tag{12}$$

*in Algorithms 1 for any integer $t \ge 1$. For every $t \ge 1$,*

$$\mathbb{E}[\|d_t - \nabla J_H(\theta_t)\|] \le (1-\eta)^t\frac{\sigma}{\sqrt{M_{init}}} + \sqrt{\eta}\frac{\sigma}{\sqrt{M}} + L_h\frac{2\alpha^2}{\eta^2} + \frac{4D_g\gamma^H}{\eta} + 2D_h\gamma^H\frac{\alpha}{\eta}.$$

*Moreover, for every $T \ge 1$,*

$$\sum_{t=0}^{T-1}\mathbb{E}[\|d_t - \nabla J_H(\theta_t)\|] \le \frac{\sigma}{\sqrt{M_{init}}\eta} + \frac{\sqrt{\eta}\sigma T}{\sqrt{M}} + \frac{2\alpha^2 L_h T}{\eta^2} + \frac{4D_g\gamma^H T}{\eta} + \frac{2\alpha D_h\gamma^H T}{\eta}.$$

*where $D_g$ and $D_h$ are defined in Proposition 2.3.*

*Proof.* Let us define

$$\hat{e}_t := d_t - \nabla J_H(\theta_t),$$
$$e_t := g_t - \nabla J_H(\widetilde{\theta}_t),$$
$$S_t := \nabla J_H(\theta_{t-1}) - \nabla J_H(\theta_t) + \nabla^2 J_H(\theta_t)(\theta_{t-1} - \theta_t),$$
$$\bar{S}_t := \nabla J(\theta_{t-1}) - \nabla J(\theta_t) + \nabla^2 J(\theta_t)(\theta_{t-1} - \theta_t),$$
$$Z_t := \nabla J_H(\widetilde{\theta}_t) - \nabla J_H(\theta_t) + \nabla^2 J_H(\theta_t)(\widetilde{\theta}_t - \theta_t),$$
$$\bar{Z}_t := \nabla J(\widetilde{\theta}_t) - \nabla J(\theta_t) + \nabla^2 J(\theta_t)(\widetilde{\theta}_t - \theta_t).$$

Using triangle inequality and Proposition 2.3-(2),(3) (or Proposition 2.4-(2),(3)), we get

$$\|S_t\| \leq L_h \|\theta_t - \theta_{t-1}\|^2 + \|S_t - \bar{S}_t\| = L_h \alpha^2 + 2D_g \gamma^H + D_h \gamma^H \alpha. \tag{13}$$

Similarly,

$$\|Z_t\| \leq L_h \frac{(1-\eta)^2 \alpha^2}{\eta^2} + 2D_g \gamma^H + D_h \gamma^H \frac{(1-\eta)\alpha}{\eta}. \tag{14}$$

Applying the update rule for $d_t$ Algorithm 1:

$$d_t = (1-\eta)d_{t-1} + \eta g_t, \tag{15}$$

connecting this with notation for $S_t$ and $Z_t$, and using $\widetilde{\theta}_t = \theta_t + \frac{1-\eta}{\eta}(\theta_t - \theta_{t-1})$, we derive the recursion

$$\hat{e}_t = d_t - \nabla J_H(\theta_t) = (1-\eta)d_{t-1} + \eta g_t - \nabla J_H(\theta_t)$$
$$= (1-\eta)\hat{e}_{t-1} + \eta e_t + (1-\eta)S_t + \eta Z_t.$$

We can rewrite unroll the recursion and get

$$\hat{e}_t = (1-\eta)^t(d_0 - \nabla J_H(\theta_0)) + \eta \sum_{\tau=0}^{t-1}(1-\eta)^{t-\tau-1}e_{\tau+1} + (1-\eta)\sum_{\tau=0}^{t-1}(1-\eta)^{t-\tau-1}S_{\tau+1} + \eta\sum_{\tau=0}^{t-1}(1-\eta)^{t-\tau-1}Z_{\tau+1}.$$

Now, we estimate the expectation of the norm. Using triangle inequality,

$$\mathbb{E}[\|\hat{e}_t\|] \leq (1-\eta)^t \mathbb{E}[\|d_0 - \nabla J_H(\theta_0)\|] + \eta\mathbb{E}\left[\left\|\sum_{\tau=0}^{t-1}(1-\eta)^{t-\tau-1}e_{\tau+1}\right\|\right] +$$

$$+ (1-\eta)\mathbb{E}\left[\left\|\sum_{\tau=0}^{t-1}(1-\eta)^{t-\tau-1}S_{\tau+1}\right\|\right] + \eta\mathbb{E}\left[\left\|\sum_{\tau=0}^{t-1}(1-\eta)^{t-\tau-1}Z_{\tau+1}\right\|\right]$$

$$\leq (1-\eta)^t \frac{\sigma}{\sqrt{M_{\text{init}}}} + \left(\eta^2\mathbb{E}\left[\left\|\sum_{\tau=0}^{t-1}(1-\eta)^{t-\tau-1}e_{\tau+1}\right\|^2\right]\right)^{1/2} +$$

$$+ (1-\eta)\sum_{\tau=0}^{t-1}(1-\eta)^{t-\tau-1}\mathbb{E}[\|S_{\tau+1}\|] + \eta\sum_{\tau=0}^{t-1}(1-\eta)^{t-\tau-1}\mathbb{E}[\|Z_{\tau+1}\|],$$

where we use the Jensen's inequality , triangle inequality, and (11); Using (12), for all $i > j$, $\mathbb{E}[\langle e_i, e_j \rangle] = \mathbb{E}[\mathbb{E}_i[\langle e_i, e_j \rangle]] = \mathbb{E}[\langle \mathbb{E}_i[e_i], e_j \rangle] = 0$, and we get

$$\mathbb{E}[\|\hat{e}_t\|] \leq (1-\eta)^t \frac{\sigma}{\sqrt{M_{\text{init}}}} + \left(\eta^2\sum_{\tau=0}^{t-1}(1-\eta)^{2(t-\tau-1)}\frac{\sigma^2}{M}\right)^{1/2} +$$

$$+ (1-\eta)\sum_{\tau=0}^{t-1}(1-\eta)^{t-\tau-1}\mathbb{E}[\|S_{\tau+1}\|] + \eta\sum_{\tau=0}^{t-1}(1-\eta)^{t-\tau-1}\mathbb{E}[\|Z_{\tau+1}\|]$$

$$\leq (1-\eta)^t \frac{\sigma}{\sqrt{M_{\text{init}}}} + \sqrt{\eta}\frac{\sigma}{\sqrt{M}} + (1-\eta)\sum_{\tau=0}^{t-1}(1-\eta)^{t-\tau-1}\mathbb{E}[\|S_{\tau+1}\|] + \eta\sum_{\tau=0}^{t-1}(1-\eta)^{t-\tau-1}\mathbb{E}[\|Z_{\tau+1}\|],$$

where we use $\sum_{\tau=0}^{t-1}(1-\eta)^{2(t-\tau-1)} \leq \frac{1}{\eta}$. Using (13) and (14),

$$
\begin{aligned}
\mathbb{E}[\|\hat{e}_t\|] &\leq (1-\eta)^t \frac{\sigma}{\sqrt{M_{\text{init}}}} + \sqrt{\eta}\frac{\sigma}{\sqrt{M}} \\
&\quad + (1-\eta)\sum_{\tau=0}^{t-1}(1-\eta)^{t-\tau-1}\left(L_h\alpha^2 + 2D_g\gamma^H + D_h\gamma^H\alpha\right) \\
&\quad + \eta\sum_{\tau=0}^{t-1}(1-\eta)^{t-\tau-1}\left(L_h\frac{(1-\eta)^2\alpha^2}{\eta^2} + 2D_g\gamma^H + D_h\gamma^H\frac{(1-\eta)\alpha}{\eta}\right) \\
&\leq (1-\eta)^t \frac{\sigma}{\sqrt{M_{\text{init}}}} + \sqrt{\eta}\frac{\sigma}{\sqrt{M}} + L_h\frac{2\alpha^2}{\eta^2} + \frac{4D_g\gamma^H}{\eta} + 2D_h\gamma^H\frac{\alpha}{\eta},
\end{aligned}
$$

where we use $\sum_{\tau=0}^{t-1}(1-\eta)^{t-\tau-1} \leq \frac{1}{\eta}$ and $0 < \eta < 1$. It left sum the inequality and use $\sum_{t=0}^{T-1}(1-\eta)^t \leq \frac{1}{\eta}$ to get

$$
\sum_{t=0}^{T-1}\mathbb{E}[\|\hat{e}_t\|] \leq \frac{\sigma}{\sqrt{M_{\text{init}}}\eta} + \sqrt{\eta}\frac{\sigma}{\sqrt{M}}T + \frac{2\alpha^2 L_h T}{\eta^2} + \frac{4D_g\gamma^H T}{\eta} + \frac{2\alpha D_h\gamma^H T}{\eta}.
$$

$\square$

**Theorem 4.1.** *Let Assumptions 2.1 and 2.2 hold. Consider* Rennala NIGT *(Algorithm 1 and Algorithm 2) in the homogeneous setup, or* Malenia NIGT *(Algorithms 1 and Algorithm 3) in the heterogeneous setup. Let* $\eta = \min\left\{\frac{M\varepsilon^2}{64\sigma^2}, \frac{1}{2}\right\}$, $\alpha = \min\left\{\frac{\varepsilon}{8L_g}, \frac{\eta\sqrt{\varepsilon}}{4\sqrt{L_h}}\right\}$, $H = \max\left\{\log_\gamma\left(\frac{\varepsilon\eta}{64\max\{D_g,\alpha D_h\}}\right), 1\right\} = \tilde{\mathcal{O}}(1/(1-\gamma))$. *Let* $\bar{\theta}_T$ *be a uniformly sampled iterate from* $\{\theta_0, \cdots, \theta_{T-1}\}$. *Then* $\mathbb{E}\left\|\nabla J(\bar{\theta}_T)\right\| \leq \varepsilon$ *after* $T = \mathcal{O}\left(\frac{L_g\Delta}{\varepsilon^2} + \frac{\sqrt{L_h}\Delta}{\varepsilon^{3/2}} + \frac{\sigma}{\sqrt{M_{init}}\varepsilon} + \frac{\sigma^3}{M\sqrt{M_{init}}\varepsilon^3} + \frac{\sigma^2\sqrt{L_h}\Delta}{M\varepsilon^{7/2}}\right)$ *global iterations.*

*Proof.* We rely on Lemma D.2, which requires $d_0$ and $\{g_t\}$ to satisfy (11) and (12). For Rennala NIGT with Algorithm 2, this condition holds because $\mathbb{E}[\bar{g}] = \frac{1}{M}\sum_{j=1}^{M}\mathbb{E}[g_H(\bar{\tau}_j, \theta)] = \nabla J_H(\theta)$ for all $\theta \in \mathbb{R}^d$, where $\bar{\tau}_j \sim p(\cdot|\pi_\theta)$ are i.i.d. trajectories sampled by the agents. (For instance, if agent 1 is the first to return a stochastic gradient, then $\bar{\tau}_1 \equiv \tau_{1,1}$. If agent 3 is the next, then $\bar{\tau}_2 \equiv \tau_{3,1}$. If agent 1 returns again, then $\bar{\tau}_3 \equiv \tau_{1,2}$, and so on.) Moreover,

$$
\begin{aligned}
\mathbb{E}\left[\|\bar{g} - \nabla J_H(\theta)\|^2\right] &= \mathbb{E}\left[\left\|\frac{1}{M}\sum_{j=1}^{M}g_H(\bar{\tau}_j, \theta) - \nabla J_H(\theta)\right\|^2\right] \\
&= \frac{1}{M^2}\sum_{j=1}^{M}\mathbb{E}\left[\|g_H(\bar{\tau}_j, \theta) - \nabla J_H(\theta)\|^2\right] \leq \frac{\sigma^2}{M}
\end{aligned}
$$

for all $\theta \in \mathbb{R}^d$ due to the unbiasedness, the i.i.d. nature of the trajectories, and Proposition 2.3-(4).

In the case of Malenia NIGT with Algorithm 3, (11) and (12) also hold since $\mathbb{E}\left[\frac{1}{n}\sum_{i=1}^{n}\frac{\bar{g}_i}{M_i}\right] = \frac{1}{n}\sum_{i=1}^{n}\frac{1}{M_i}\sum_{j=1}^{M_i}\mathbb{E}[g_{i,H}(\tau_{i,j}, \theta)] = \frac{1}{n}\sum_{i=1}^{n}\nabla J_{i,H}(\theta) = \nabla J_H(\theta)$, and

$$
\begin{aligned}
\mathbb{E}\left[\left\|\frac{1}{n}\sum_{i=1}^{n}\frac{1}{M_i}\sum_{j=1}^{M_i}g_{i,H}(\tau_{i,j}, \theta) - \nabla J_H(\theta)\right\|^2\right] &= \mathbb{E}\left[\left\|\frac{1}{n}\sum_{i=1}^{n}\left(\frac{1}{M_i}\sum_{j=1}^{M_i}g_{i,H}(\tau_{i,j}, \theta) - \nabla J_{i,H}(\theta)\right)\right\|^2\right] \\
&= \frac{1}{n^2}\sum_{i=1}^{n}\frac{1}{M_i^2}\sum_{j=1}^{M_i}\mathbb{E}\left[\|g_{i,H}(\tau_{i,j}, \theta) - \nabla J_{i,H}(\theta)\|^2\right] \leq \frac{1}{n^2}\sum_{i=1}^{n}\frac{1}{M_i}\sigma^2.
\end{aligned}
$$

for all $\theta \in \mathbb{R}^d$, where we use $\mathbb{E}[g_{i,H}(\tau_{i,j}, \theta)] = \nabla J_{i,H}(\theta)$ for $i \in [n], j \geq 1$, independence, and Proposition 2.4-(4). It left to use the exit loop condition of Algorithm 3, which ensure that

$(1/n \sum_{i=1}^{n} 1/M_i)^{-1} \geq M/n$ and

$$\mathbb{E} \left[ \left\| \frac{1}{n} \sum_{i=1}^{n} \frac{1}{M_i} \sum_{j=1}^{M_i} g_{i,H}(\tau_{i,j}, \theta) - \nabla J_H(\theta) \right\|^2 \right] \leq \frac{\sigma^2}{M}.$$

Using (10) from Lemma D.1 and summing it for $t = 0, \ldots, T-1$,

$$\frac{1}{T} \mathbb{E}[J^* - J(\theta_T)] \leq \frac{1}{T} (J^* - J(\theta_0)) - \alpha \frac{1}{T} \sum_{t=0}^{T-1} \mathbb{E}[\|\nabla J(\theta_t)\|] + 2\alpha \frac{1}{T} \sum_{t=0}^{T-1} \mathbb{E}[\|d_t - \nabla J(\theta_t)\|] + \frac{L_g \alpha^2}{2}.$$

Due to $(J^* - J(\theta_T)) \geq 0$ and Lemma D.2,

$$\frac{1}{T} \sum_{t=0}^{T-1} \mathbb{E}[\|\nabla J(\theta_t)\|] \leq \frac{\Delta}{\alpha T} + \frac{L_g \alpha}{2} + \left( \frac{2\sigma}{\sqrt{M_{\text{init}}} T \eta} + 2\sqrt{\eta} \frac{\sigma}{\sqrt{M}} + \frac{4\alpha^2 L_h}{\eta^2} + \frac{8 D_g \gamma^H}{\eta} + \frac{4\alpha D_h \gamma^H}{\eta} \right)$$

Choosing $H = \max \left\{ \log_\gamma \left( \frac{\varepsilon \eta}{64 \max\{D_g, \alpha D_h\}} \right), 1 \right\}$, we get $\frac{8 D_g \gamma^H}{\eta} + \frac{4\alpha D_h \gamma^H}{\eta} \leq \frac{\varepsilon}{4}$ and

$$\frac{1}{T} \sum_{t=0}^{T-1} \mathbb{E}[\|\nabla J(\theta_t)\|] \leq \frac{\Delta}{\alpha T} + \frac{L_g \alpha}{2} + \frac{2\sigma}{\sqrt{M_{\text{init}}} T \eta} + 2\sqrt{\eta} \frac{\sigma}{\sqrt{M}} + \frac{4\alpha^2 L_h}{\eta^2} + \frac{\varepsilon}{4}$$

Now, we set $\alpha = \min \left\{ \frac{\varepsilon}{8 L_g}, \frac{\eta \sqrt{\varepsilon}}{4 \sqrt{L_h}} \right\}$ to get

$$\frac{1}{T} \sum_{t=0}^{T-1} \mathbb{E}[\|\nabla J(\theta_t)\|] \leq \frac{\Delta}{\alpha T} + \frac{2\sigma}{\sqrt{M_{\text{init}}} T \eta} + 2\sqrt{\eta} \frac{\sigma}{\sqrt{M}} + \frac{\varepsilon}{2}.$$

We choose $\eta = \min \left\{ \frac{M \varepsilon^2}{64 \sigma^2}, \frac{1}{2} \right\}$ :

$$\frac{1}{T} \sum_{t=0}^{T-1} \mathbb{E}[\|\nabla J(\theta_t)\|] \leq \frac{\Delta}{\alpha T} + \frac{2\sigma}{\sqrt{M_{\text{init}}} T \eta} + \frac{3\varepsilon}{4}.$$

Thus, the method converges after

$$T = \mathcal{O} \left( \frac{\Delta}{\varepsilon \alpha} + \frac{\sigma}{\sqrt{M_{\text{init}}} \varepsilon \eta} \right)$$

$$= \mathcal{O} \left( \frac{L_g \Delta}{\varepsilon^2} + \frac{\sqrt{L_h} \Delta}{\varepsilon^{3/2} \eta} + \frac{\sigma}{\sqrt{M_{\text{init}}} \varepsilon \eta} \right)$$

$$= \mathcal{O} \left( \frac{L_g \Delta}{\varepsilon^2} + \frac{\sqrt{L_h} \Delta}{\varepsilon^{3/2}} + \frac{\sigma^2 \sqrt{L_h} \Delta}{M \varepsilon^{7/2}} + \frac{\sigma}{\sqrt{M_{\text{init}}} \varepsilon} + \frac{\sigma^3}{M \sqrt{M_{\text{init}}} \varepsilon^3} \right)$$

global iterations. $\qquad \square$

# E  TIME COMPLEXITY

## E.1  TIME COMPLEXITY OF Rennala NIGT

**Theorem 4.2.** *Consider the results and assumptions of Theorem 4.1. Additionally, consider that Assumption 2.5 holds with $\kappa = 0$ (i.e., communication is free). Taking $M_{init} = \max \left\{ \lceil \frac{\sigma^2}{\varepsilon^2} \rceil, 1 \right\}$ and $M = \max \left\{ \lceil \left( \frac{\sigma^2}{\varepsilon^2} + \frac{\sigma^2 \sqrt{L_h} \Delta}{\varepsilon^{7/2}} \right) / \left( \frac{L_g \Delta}{\varepsilon^2} + \frac{\sqrt{L_h} \Delta}{\varepsilon^{3/2}} \right) \rceil, 1 \right\}$, the time required to find an $\varepsilon$–stationary point by* Rennala NIGT *(Algorithms 1 and 2) is*

$$\tilde{\mathcal{O}} \left( \frac{1}{1-\gamma} \min_{m \in [n]} \left[ \left( \frac{1}{m} \sum_{i=1}^{m} \frac{1}{h_i} \right)^{-1} \left( \frac{L_g \Delta}{\varepsilon^2} + \frac{\sqrt{L_h} \Delta}{\varepsilon^{3/2}} + \frac{\sigma^2}{m \varepsilon^2} + \frac{\sigma^2 \sqrt{L_h} \Delta}{m \varepsilon^{7/2}} \right) \right] \right). \qquad (6)$$

*Proof.* With our choice of $M$ and $M_{\text{init}}$, the number of global iterations to get an $\varepsilon$–stationary point is

$$T = \mathcal{O}\left(\frac{L_g\Delta}{\varepsilon^2} + \frac{\sqrt{L_h}\Delta}{\varepsilon^{3/2}} + \frac{\sigma^2\sqrt{L_h}\Delta}{M\varepsilon^{7/2}} + \frac{\sigma}{\sqrt{M_{\text{init}}}\varepsilon} + \frac{\sigma^3}{M\sqrt{M_{\text{init}}}\varepsilon^3}\right)$$

$$= \mathcal{O}\left(\frac{L_g\Delta}{\varepsilon^2} + \frac{\sqrt{L_h}\Delta}{\varepsilon^{3/2}} + \frac{\sigma^2}{M\varepsilon^2} + \frac{\sigma^2\sqrt{L_h}\Delta}{M\varepsilon^{7/2}}\right) = \mathcal{O}\left(\frac{L_g\Delta}{\varepsilon^2} + \frac{\sqrt{L_h}\Delta}{\varepsilon^{3/2}}\right).$$

Notice that the time requires to collect $M$ stochastic gradients in Algorithm 2 is

$$\mathcal{O}\left(\min_{m\in[n]}\left[\left(\sum_{i=1}^{m}\frac{1}{h_i}\right)^{-1}(M+m)\right]\right).$$

with $\kappa = 0$ due to Lemma C.1. At the beginning, the algorithm collects $M_{\text{init}}$ stochastic gradients that takes $\mathcal{O}\left(\min_{m\in[n]}\left[\left(\sum_{i=1}^{m}\frac{1}{h_i}\right)^{-1}(M_{\text{init}}+m)\right]\right)$ seconds. Then, in each iteration the agents collect $M$ stochastic gradients, which takes $\mathcal{O}\left(\min_{m\in[n]}\left[\left(\sum_{i=1}^{m}\frac{1}{h_i}\right)^{-1}(M+m)\right]\right)$ seconds. Thus, after $T$ iterations, the total time to find an $\varepsilon$–stationary point is

$$\mathcal{O}\left(T\times\min_{m\in[n]}\left[\left(\sum_{i=1}^{m}\frac{1}{h_i}\right)^{-1}(M+m)\right] + \min_{m\in[n]}\left[\left(\sum_{i=1}^{m}\frac{1}{h_i}\right)^{-1}(M_{\text{init}}+m)\right]\right)$$

$$= \mathcal{O}\left(\min_{m\in[n]}\left[\left(\frac{1}{m}\sum_{i=1}^{m}\frac{1}{h_i}\right)^{-1}\left(\frac{L_g\Delta}{\varepsilon^2} + \frac{\sqrt{L_h}\Delta}{\varepsilon^{3/2}} + \frac{\sigma^2}{m\varepsilon^2} + \frac{\sigma^2\sqrt{L_h}\Delta}{m\varepsilon^{7/2}}\right)\right]\right)$$

$$+ \mathcal{O}\left(\min_{m\in[n]}\left[\left(\frac{1}{m}\sum_{i=1}^{m}\frac{1}{h_i}\right)^{-1}\left(\frac{\sigma^2}{m\varepsilon^2} + 1\right)\right]\right)$$

$$= \mathcal{O}\left(\min_{m\in[n]}\left[\left(\frac{1}{m}\sum_{i=1}^{m}\frac{1}{h_i}\right)^{-1}\left(\frac{L_g\Delta}{\varepsilon^2} + \frac{\sqrt{L_h}\Delta}{\varepsilon^{3/2}} + \frac{\sigma^2}{m\varepsilon^2} + \frac{\sigma^2\sqrt{L_h}\Delta}{m\varepsilon^{7/2}}\right)\right]\right).$$

It is left to substitute $h_i = \dot{h}_i \times H$ and recall that $H = \tilde{\mathcal{O}}\left(\frac{1}{1-\gamma}\right)$ to get (6). □

**Theorem 4.3.** *Consider the results and assumptions of Theorem 4.1. Additionally, consider that computing a single stochastic policy gradient $g_H$ on agent $i$ in iteration $t$ requires at most $\dot{h}_{t,i} \times H$ seconds. Taking $M_{init} = \max\left\{\left\lceil\frac{\sigma^2}{\varepsilon^2}\right\rceil, 1\right\}$ and $M = \max\left\{\left\lceil\left(\frac{\sigma^2}{\varepsilon^2} + \frac{\sigma^2\sqrt{L_h}\Delta}{\varepsilon^{7/2}}\right)/\left(\frac{L_g\Delta}{\varepsilon^2} + \frac{\sqrt{L_h}\Delta}{\varepsilon^{3/2}}\right)\right\rceil, 1\right\}$, the time required to find an $\varepsilon$–stationary point by* Rennala NIGT *(Algorithms 1 and 2) is $\tilde{\mathcal{O}}\left(\frac{1}{1-\gamma}\sum_{t=1}^{T}\min_{m\in[n]}\left[\left(\frac{1}{m}\sum_{i=1}^{m}\frac{1}{h_{t,\pi_{t,i}}}\right)^{-1}\left(\frac{M}{m} + 1\right)\right] + \frac{1}{1-\gamma}\min_{m\in[n]}\left[\left(\frac{1}{m}\sum_{i=1}^{m}\frac{1}{h_{t,\pi_{0,i}}}\right)^{-1}\left(\frac{M_{init}}{m} + 1\right)\right]\right)$, where $T = \mathcal{O}\left(\frac{L_g\Delta}{\varepsilon^2} + \frac{\sqrt{L_h}\Delta}{\varepsilon^{3/2}}\right)$ and $\pi_{t,i}$ is a permutation such that $\dot{h}_{t,\pi_{t,1}} \leq \cdots \leq \dot{h}_{t,\pi_{t,n}}$.*

*Proof.* The proof is the same as in Theorem 4.2. One should only notice that the total time complexity is

$$\tilde{\mathcal{O}}\left(\sum_{t=1}^{T}\min_{m\in[n]}\left[\left(\frac{1}{m}\sum_{i=1}^{m}\frac{1}{h_{t,\pi_{t,i}}}\right)^{-1}\left(\frac{M}{m} + 1\right)\right] + \min_{m\in[n]}\left[\left(\frac{1}{m}\sum_{i=1}^{m}\frac{1}{h_{t,\pi_{0,i}}}\right)^{-1}\left(\frac{M_{\text{init}}}{m} + 1\right)\right]\right)$$

because Algorithm 2 requires at most $\min_{m\in[n]}\left[\left(\frac{1}{m}\sum_{i=1}^{m}\frac{1}{h_{t,\pi_{t,i}}}\right)^{-1}\left(\frac{M}{m} + 1\right)\right]$ seconds to collect a batch of size $M$ in $t^{\text{th}}$ iteration, and $\min_{m\in[n]}\left[\left(\frac{1}{m}\sum_{i=1}^{m}\frac{1}{h_{t,\pi_{0,i}}}\right)^{-1}\left(\frac{M_{\text{init}}}{m} + 1\right)\right]$ seconds to collect a batch of size $M_{\text{init}}$ before the loop. It is left to substitute $h_{i,j} = \dot{h}_{i,j} \times H$. □

**Theorem 4.4.** *Consider the results and assumptions of Theorem 4.1. Additionally, consider that Assumption 2.5 holds. Taking $M_{init} = \max\left\{\left\lceil\frac{\sigma^2}{\varepsilon^2}\right\rceil, 1\right\}$ and $M = \max\left\{\left\lceil\left(\frac{\sigma^2}{\varepsilon^2} + \frac{\sigma^2\sqrt{L_h}\Delta}{\varepsilon^{7/2}}\right)/\left(\frac{L_g\Delta}{\varepsilon^2} + \frac{\sqrt{L_h}\Delta}{\varepsilon^{3/2}}\right)\right\rceil, 1\right\}$, the time required to find an $\varepsilon$–stationary point by* Rennala NIGT *(Algorithms 1 and 2) is*

$$\tilde{\mathcal{O}}\left(\kappa\left(\frac{L_g\Delta}{\varepsilon^2} + \frac{\sqrt{L_h}\Delta}{\varepsilon^{3/2}}\right) + \frac{1}{1-\gamma}\min_{m\in[n]}\left[\left(\frac{1}{m}\sum_{i=1}^{m}\frac{1}{h_i}\right)^{-1}\left(\frac{L_g\Delta}{\varepsilon^2} + \frac{\sqrt{L_h}\Delta}{\varepsilon^{3/2}} + \frac{\sigma^2}{m\varepsilon^2} + \frac{\sigma^2\sqrt{L_h}\Delta}{m\varepsilon^{7/2}}\right)\right]\right).$$

*Proof.* The second term in the complexity is proved in Theorem 4.2. However, Theorem 4.2 does not take into account the first communication term. Using the same reasoning, after $T$ iterations, the total time to find an $\varepsilon$–stationary point is

$$\mathcal{O}\left(T\times\kappa + T\times\min_{m\in[n]}\left[\left(\sum_{i=1}^{m}\frac{1}{h_i}\right)^{-1}(M+m)\right] + \kappa + \min_{m\in[n]}\left[\left(\sum_{i=1}^{m}\frac{1}{h_i}\right)^{-1}(M_{init}+m)\right]\right)$$

$$= \mathcal{O}\left(\kappa\left(\frac{L_g\Delta}{\varepsilon^2} + \frac{\sqrt{L_h}\Delta}{\varepsilon^{3/2}}\right) + \min_{m\in[n]}\left[\left(\frac{1}{m}\sum_{i=1}^{m}\frac{1}{h_i}\right)^{-1}\left(\frac{L_g\Delta}{\varepsilon^2} + \frac{\sqrt{L_h}\Delta}{\varepsilon^{3/2}} + \frac{\sigma^2}{m\varepsilon^2} + \frac{\sigma^2\sqrt{L_h}\Delta}{m\varepsilon^{7/2}}\right)\right]\right).$$

It is left to substitute $h_i = \dot{h}_i \times H$ and recall that $H = \tilde{\mathcal{O}}\left(\frac{1}{1-\gamma}\right).$ □

### E.2 TIME COMPLEXITY OF Rennala NIGT UNDER ASSUMPTION 7.1

**Theorem 7.2.** *Consider the results and assumptions of Theorem 4.1. Additionally, consider that Assumption 7.1 holds. Taking $M_{init} = \max\left\{\left\lceil\frac{\sigma^2}{\varepsilon^2}\right\rceil, 1\right\}$ and $M = \max\left\{\left\lceil\left(\frac{\sigma^2}{\varepsilon^2} + \frac{\sigma^2\sqrt{L_h}\Delta}{\varepsilon^{7/2}}\right)/\left(\frac{L_g\Delta}{\varepsilon^2} + \frac{\sqrt{L_h}\Delta}{\varepsilon^{3/2}}\right)\right\rceil, 1\right\}$, the time required to find an $\varepsilon$–stationary point by* Rennala NIGT *(Algorithms 1 and 2) is $t_T$ seconds, where $T = \mathcal{O}\left(\frac{L_g\Delta}{\varepsilon^2} + \frac{\sqrt{L_h}\Delta}{\varepsilon^{3/2}}\right)$ and the sequence $t_k$ is defined recursively:*

$$t_k = \min\left\{t : \sum_{i=1}^{n} N_i(t_{k-1}, t) \geq M\right\} \qquad \forall k \geq 1 \tag{8}$$

*with $t_0 = \min\left\{t : \sum_{i=1}^{n} N_i(0, t) \geq M_{init}\right\}.$*

*Proof.* With our choice of $M$ and $M_{init}$, the number of global iterations to get an $\varepsilon$–stationary point is

$$T = \mathcal{O}\left(\frac{L_g\Delta}{\varepsilon^2} + \frac{\sqrt{L_h}\Delta}{\varepsilon^{3/2}}\right).$$

At the beginning, the algorithm collects $M_{init}$ stochastic gradients that takes

$$t_0 = \min\left\{t : \sum_{i=1}^{n} N_i(0, t) \geq M_{init}\right\}$$

seconds because $N_i(0, t)$ is the number of calculated gradients in agent $i$ and they work in parallel. Then, in each iteration $k$ the agents collect $M$ stochastic gradients. The first iteration finishes after at most

$$t_1 = \min\left\{t : \sum_{i=1}^{n} N_i(t_0, t) \geq M\right\}$$

seconds. Similarly, the $k^{\text{th}}$ iteration finishes after at most

$$t_k = \min\left\{t : \sum_{i=1}^{n} N_i(t_{k-1}, t) \geq M\right\}$$

seconds. Thus, $T$ iterations finish after at most $t_T$ seconds. □

### E.3 TIME COMPLEXITY OF Malenia NIGT

**Theorem 5.1.** *Consider the results and assumptions of Theorem 4.1. Additionally, consider that Assumption 2.5 holds. Taking $M_{init} = \max\left\{\lceil\frac{\sigma^2}{\varepsilon^2}\rceil, 1\right\}$ and $M = \max\left\{\lceil\left(\frac{\sigma^2}{\varepsilon^2} + \frac{\sigma^2\sqrt{L_h}\Delta}{\varepsilon^{7/2}}\right)/\left(\frac{L_g\Delta}{\varepsilon^2} + \frac{\sqrt{L_h}\Delta}{\varepsilon^{3/2}}\right)\rceil, 1\right\}$, the time required to find an $\varepsilon$–stationary point by Malenia NIGT (Algorithms 1 and 3) is*

$$\tilde{\mathcal{O}}\left(\kappa\left(\frac{L_g\Delta}{\varepsilon^2} + \frac{\sqrt{L_h}\Delta}{\varepsilon^{3/2}}\right) + \frac{1}{1-\gamma}\left[\dot{h}_n\left(\frac{L_g\Delta}{\varepsilon^2} + \frac{\sqrt{L_h}\Delta}{\varepsilon^{3/2}}\right) + \left(\frac{1}{n}\sum_{i=1}^{n}\dot{h}_i\right)\left(\frac{\sigma^2}{n\varepsilon^2} + \frac{\sigma^2\sqrt{L_h}\Delta}{n\varepsilon^{7/2}}\right)\right]\right).$$

*Proof.* Similarly to the proof of Theorem 4.2, with our choice of $M$ and $M_{init}$,

$$T = \mathcal{O}\left(\frac{L_g\Delta}{\varepsilon^2} + \frac{\sqrt{L_h}\Delta}{\varepsilon^{3/2}}\right).$$

At the beginning, the algorithm collects $M_{init}$ stochastic gradients that takes $\mathcal{O}\left(\kappa + h_n + \left(\frac{1}{n}\sum_{i=1}^{n}h_i\right)\frac{M_{init}}{n}\right)$ seconds due to Lemma C.2. Then, in each iteration the agents collect $M$ stochastic gradients, which takes $\mathcal{O}\left(\kappa + h_n + \left(\frac{1}{n}\sum_{i=1}^{n}h_i\right)\frac{M}{n}\right)$ seconds. Thus, after $T$ iterations, the total time to find an $\varepsilon$–stationary point is

$$\mathcal{O}\left(\kappa + h_n + \left(\frac{1}{n}\sum_{i=1}^{n}h_i\right)\frac{M_{init}}{n} + T\times\left(\kappa + h_n + \left(\frac{1}{n}\sum_{i=1}^{n}h_i\right)\frac{M}{n}\right)\right)$$

$$= \mathcal{O}\left(\left(\frac{L_g\Delta}{\varepsilon^2} + \frac{\sqrt{L_h}\Delta}{\varepsilon^{3/2}}\right)(\kappa + h_n) + \left(\frac{1}{n}\sum_{i=1}^{n}h_i\right)\frac{\sigma^2}{n\varepsilon^2} + \left(\left(\frac{1}{n}\sum_{i=1}^{n}h_i\right)\left(\frac{\sigma^2}{n\varepsilon^2} + \frac{\sigma^2\sqrt{L_h}\Delta}{n\varepsilon^{7/2}}\right)\right)\right)$$

$$= \mathcal{O}\left((\kappa + h_n)\left(\frac{L_g\Delta}{\varepsilon^2} + \frac{\sqrt{L_h}\Delta}{\varepsilon^{3/2}}\right) + \left(\frac{1}{n}\sum_{i=1}^{n}h_i\right)\left(\frac{\sigma^2}{n\varepsilon^2} + \frac{\sigma^2\sqrt{L_h}\Delta}{n\varepsilon^{7/2}}\right)\right),$$

where we substitute our choice of $M$ and $M_{init}$, and use $T = \mathcal{O}\left(\frac{L_g\Delta}{\varepsilon^2} + \frac{\sqrt{L_h}\Delta}{\varepsilon^{3/2}}\right)$. It is left to substitute $h_i = \dot{h}_i \times H$ and recall that $H = \tilde{\mathcal{O}}\left(\frac{1}{1-\gamma}\right)$ $\qquad\square$

### E.4 TIME COMPLEXITY OF Malenia NIGT UNDER ASSUMPTION 7.1

**Theorem 7.3.** *Consider the results and assumptions of Theorem 4.1. Additionally, consider that Assumption 7.1 holds. Taking $M_{init} = \max\left\{\lceil\frac{\sigma^2}{\varepsilon^2}\rceil, 1\right\}$ and $M = \max\left\{\lceil\left(\frac{\sigma^2}{\varepsilon^2} + \frac{\sigma^2\sqrt{L_h}\Delta}{\varepsilon^{7/2}}\right)/\left(\frac{L_g\Delta}{\varepsilon^2} + \frac{\sqrt{L_h}\Delta}{\varepsilon^{3/2}}\right)\rceil, 1\right\}$, the time required to find an $\varepsilon$–stationary point by Malenia NIGT (Algorithms 1 and 3) is $t_T$ seconds, where $T = \mathcal{O}\left(\frac{L_g\Delta}{\varepsilon^2} + \frac{\sqrt{L_h}\Delta}{\varepsilon^{3/2}}\right)$ and the sequence $t_k$ is defined recursively:*

$$t_k = \min\left\{t : \left(\frac{1}{n}\sum_{i=1}^{n}\frac{1}{N_i(t_{k-1}, t)}\right)^{-1} \geq \frac{M}{n}\right\} \qquad \forall k \geq 1 \tag{9}$$

*with $t_0 = \min\left\{t : \left(\frac{1}{n}\sum_{i=1}^{n}\frac{1}{N_i(0, t)}\right)^{-1} \geq \frac{M_{init}}{n}\right\}$.*

*Proof.* Similarly to the proof of Theorem 4.2, with our choice of $M$ and $M_{init}$,

$$T = \mathcal{O}\left(\frac{L_g\Delta}{\varepsilon^2} + \frac{\sqrt{L_h}\Delta}{\varepsilon^{3/2}}\right).$$

At the beginning, the algorithm collects $M_{init}$ stochastic gradients. According to Algorithm 3, we wait for the moment when

$$\left(\frac{1}{n}\sum_{i=1}^{n}\frac{1}{M_i}\right)^{-1} \geq \frac{M_{init}}{n}.$$

Since $M_i = N_i(0, t)$, the initial phase finishes after at most

$$t_0 = \min\left\{ t \; : \; \left( \frac{1}{n} \sum_{i=1}^{n} \frac{1}{N_i(0, t)} \right)^{-1} \geq \frac{M_{\text{init}}}{n} \right\}$$

seconds. Similarly, the $k^{\text{th}}$ iteration finishes after at most

$$t_k = \min\left\{ t \; : \; \left( \frac{1}{n} \sum_{i=1}^{n} \frac{1}{N_i(t_{k-1}, t)} \right)^{-1} \geq \frac{M}{n} \right\}$$

seconds. Thus, $T$ iterations finish after at most $t_T$ seconds. $\qquad\square$

## F  GLOBAL CONVERGENCE

**Lemma F.1** (Relaxed weak gradient domination (Ding et al. (2022)))**.** *Let Assumption 2.1 and Assumptions 2.1 and 4.6. from (Ding et al., 2022) hold. Then,*

$$\varepsilon' + \|\nabla J(\theta)\| \geq \sqrt{2\mu}(J^* - J(\theta)),$$

*for all $\theta \in \mathbb{R}^d$, where $\varepsilon' := \frac{\mu_F \sqrt{\varepsilon_{bias}}}{M_g(1-\gamma)}$ and $\mu := \frac{\mu_F^2}{2M_g^2}$. Parameter $\varepsilon_{bias}$ is an approximation error and $\mu_F$ is the smallest eigenvalue of the Fisher information matrix induced by the policy $\pi_\theta$ and and the initial state distribution (Ding et al., 2022).*

Typically, the parameter $\varepsilon_{bias}$ is small (Ding et al., 2022).

**Theorem A.1.** *Let Assumptions 2.1, 2.2, and Assumptions 2.1 and 4.6 from (Ding et al., 2022) hold. Consider* Rennala NIGT *(Algorithm 1 and Algorithm 2) in the homogeneous setup, or* Malenia NIGT *(Algorithms 1 and Algorithm 3) in the heterogeneous setup. Let $M_{init} = \max\left\{ \lceil \frac{\sigma^2}{\mu\varepsilon^2} \rceil, 1 \right\}$, $M = \max\left\{ \lceil \left( \frac{\sigma^2}{\mu\varepsilon^2} + \frac{\sigma^2 \sqrt{L_h}}{\mu^{7/4} \varepsilon^{5/2}} \right) / \left( \frac{L_g}{\mu\varepsilon} + \frac{\sqrt{L_h}}{\mu^{3/4}\sqrt{\varepsilon}} \right) \rceil, 1 \right\}$, $\eta = \min\left\{ \frac{M\mu\varepsilon^2}{64\sigma^2}, \frac{1}{2} \right\}$, $\alpha = \min\left\{ \frac{\sqrt{\mu\varepsilon}}{8L_g}, \frac{\eta\mu^{1/4}\sqrt{\varepsilon}}{4\sqrt{L_h}}, \frac{1}{\sqrt{2\mu}}, \frac{\varepsilon\eta\sqrt{M_{init}}}{8\sigma} \right\}$, $H = \max\left\{ \log_\gamma\left( \frac{\sqrt{\mu\varepsilon}\eta}{64\max\{D_g, \alpha D_h\}} \right), 1 \right\}$. Then $\mathbb{E}\left[ J^* - J(\theta_T) \right] \leq \varepsilon + \frac{\sqrt{2}\varepsilon'}{\sqrt{\mu}}$ after $T = \mathcal{O}\left( \left( \frac{L_g}{\mu\varepsilon} + \frac{\sqrt{L_h}}{\mu^{3/4}\sqrt{\varepsilon}} \right) \log\left( \frac{\Delta}{\varepsilon} \right) \right)$ global iterations, where $\varepsilon' := \frac{\mu_F \sqrt{\varepsilon_{bias}}}{M_g(1-\gamma)}$ and $\mu := \frac{\mu_F^2}{2M_g^2}$. Parameter $\varepsilon_{bias}$ is an approximation error and $\mu_F$ is the smallest eigenvalue of the Fisher information matrix induced by the policy $\pi_\theta$ and and the initial state distribution (Ding et al., 2022).*

*Proof.* In the proof of Theorem 4.1, we show that (11) and (12) are satisfied, and we can use Lemma D.2. Using (10) from Lemma D.1 and Lemma D.2:

$$\begin{aligned}
\mathbb{E}\left[ J^* - J(\theta_{t+1}) \right] &\leq \mathbb{E}\left[ J^* - J(\theta_t) \right] - \alpha\mathbb{E}\left[ \|\nabla J(\theta_t)\| \right] + 2\alpha\mathbb{E}\left[ \|d_t - \nabla J(\theta_t)\| \right] + \frac{L_g\alpha^2}{2} \\
&\leq \mathbb{E}\left[ J^* - J(\theta_t) \right] - \alpha\mathbb{E}\left[ \|\nabla J(\theta_t)\| \right] \\
&\quad + 2\alpha\left( (1-\eta)^t \frac{\sigma}{\sqrt{M_{\text{init}}}} + \sqrt{\eta}\frac{\sigma}{\sqrt{M}} + L_h \frac{2\alpha^2}{\eta^2} + \frac{4D_g\gamma^H}{\eta} + 2D_h\gamma^H \frac{\alpha}{\eta} \right) + \frac{L_g\alpha^2}{2}.
\end{aligned}$$

Due to Lemma F.1,

$$\begin{aligned}
\mathbb{E}\left[ J^* - J(\theta_{t+1}) \right] &\leq \left( 1 - \alpha\sqrt{2\mu} \right) \mathbb{E}\left[ J^* - J(\theta_t) \right] + \alpha\varepsilon' \\
&\quad + 2\alpha\left( (1-\eta)^t \frac{\sigma}{\sqrt{M_{\text{init}}}} + \sqrt{\eta}\frac{\sigma}{\sqrt{M}} + L_h \frac{2\alpha^2}{\eta^2} + \frac{4D_g\gamma^H}{\eta} + 2D_h\gamma^H \frac{\alpha}{\eta} \right) + \frac{L_g\alpha^2}{2}.
\end{aligned}$$

Unrolling the recursion,

$$\mathbb{E}\left[J^* - J(\theta_{t+1})\right] \leq \left(1 - \alpha\sqrt{2\mu}\right)^{t+1} \mathbb{E}\left[J^* - J(\theta_0)\right]$$

$$+ 2\alpha \sum_{i=0}^{t} \left(1 - \alpha\sqrt{2\mu}\right)^{i} \left((1-\eta)^{t-i}\frac{\sigma}{\sqrt{M_{\text{init}}}} + \sqrt{\eta}\frac{\sigma}{\sqrt{M}} + L_h\frac{2\alpha^2}{\eta^2} + \frac{4D_g\gamma^H}{\eta} + 2D_h\gamma^H\frac{\alpha}{\eta} + \varepsilon' + \frac{L_g\alpha}{2}\right)$$

$$\leq \left(1 - \alpha\sqrt{2\mu}\right)^{t+1} \mathbb{E}\left[J^* - J(\theta_0)\right]$$

$$+ 2\alpha \sum_{i=0}^{t} \left((1-\eta)^{t-i}\frac{\sigma}{\sqrt{M_{\text{init}}}}\right)$$

$$+ 2\alpha \sum_{i=0}^{t} \left(1 - \alpha\sqrt{2\mu}\right)^{i} \left(\sqrt{\eta}\frac{\sigma}{\sqrt{M}} + L_h\frac{2\alpha^2}{\eta^2} + \frac{4D_g\gamma^H}{\eta} + 2D_h\gamma^H\frac{\alpha}{\eta} + \varepsilon' + \frac{L_g\alpha}{2}\right).$$

Using $\sum_{i=0}^{t} \left(1 - \alpha\sqrt{2\mu}\right)^{i} \leq \frac{1}{\alpha\sqrt{2\mu}}$ for all $\alpha \leq \frac{1}{\sqrt{2\mu}}$ and $\sum_{i=0}^{t}(1-\eta)^{t-i} \leq \frac{1}{\eta}$ for all $\eta \leq 1$,

$$\mathbb{E}\left[J^* - J(\theta_{t+1})\right] \leq \left(1 - \alpha\sqrt{2\mu}\right)^{t+1} \mathbb{E}\left[J^* - J(\theta_0)\right] + \frac{2\alpha\sigma}{\eta\sqrt{M_{\text{init}}}}$$

$$+ \frac{\sqrt{2}}{\sqrt{\mu}}\left(\sqrt{\eta}\frac{\sigma}{\sqrt{M}} + L_h\frac{2\alpha^2}{\eta^2} + \frac{4D_g\gamma^H}{\eta} + 2D_h\gamma^H\frac{\alpha}{\eta} + \varepsilon' + \frac{L_g\alpha}{2}\right).$$

It is left to apply our choice of the parameters to get

$$\mathbb{E}\left[J^* - J(\theta_T)\right] \leq \left(1 - \alpha\sqrt{2\mu}\right)^{T}\Delta + \frac{3\varepsilon}{4} + \frac{\sqrt{2}\varepsilon'}{\sqrt{\mu}}$$

$$\leq \varepsilon + \frac{\sqrt{2}\varepsilon'}{\sqrt{\mu}}.$$

after $T$ iterations. □

### F.1 TIME COMPLEXITY OF Rennala NIGT

**Theorem A.2.** *Consider the results and assumptions of Theorem A.1. Additionally, consider that Assumption 2.5 holds. The time required to find an $\varepsilon$–solution up to $\sqrt{2}\varepsilon'/\sqrt{\mu}$ neighborhood by Rennala NIGT (Algorithms 1 and 2) in the homogeneous setup is*

$$\tilde{\mathcal{O}}\left(\kappa\left(\frac{L_g}{\mu\varepsilon} + \frac{\sqrt{L_h}}{\mu^{3/4}\sqrt{\varepsilon}}\right) + \frac{1}{1-\gamma}\min_{m\in[n]}\left[\left(\frac{1}{m}\sum_{i=1}^{m}\frac{1}{h_i}\right)^{-1}\left(\frac{L_g}{\mu\varepsilon} + \frac{\sqrt{L_h}}{\mu^{3/4}\sqrt{\varepsilon}} + \frac{\sigma^2}{m\mu\varepsilon^2} + \frac{\sigma^2\sqrt{L_h}}{m\mu^{7/4}\varepsilon^{5/2}}\right)\right]\right).$$

*Proof.* With our choice of $M$ and $M_{\text{init}}$, the number of global iterations is

$$T = \mathcal{O}\left(\left(\frac{L_g}{\mu\varepsilon} + \frac{\sqrt{L_h}}{\mu^{3/4}\sqrt{\varepsilon}}\right)\log\left(\frac{\Delta}{\varepsilon}\right)\right). \tag{16}$$

Notice that the time requires to collect $M$ stochastic gradients in Algorithm 2 is

$$\mathcal{O}\left(\kappa + \min_{m\in[n]}\left[\left(\sum_{i=1}^{m}\frac{1}{h_i}\right)^{-1}(M+m)\right]\right).$$

due to Lemma C.1. Thus, after $T$ iterations, the total communication time is

$$\tilde{\mathcal{O}}\left(T \times \left(\kappa + \min_{m\in[n]}\left[\left(\sum_{i=1}^{m}\frac{1}{h_i}\right)^{-1}(M+m)\right]\right) + \kappa + \min_{m\in[n]}\left[\left(\sum_{i=1}^{m}\frac{1}{h_i}\right)^{-1}(M_{\text{init}}+m)\right]\right)$$

$$= \tilde{\mathcal{O}}\left(T\kappa + T\left(\min_{m\in[n]}\left[\left(\frac{1}{m}\sum_{i=1}^{m}\frac{1}{h_i}\right)^{-1}\left(\frac{M}{m}+1\right)\right]\right) + \min_{m\in[n]}\left[\left(\frac{1}{m}\sum_{i=1}^{m}\frac{1}{h_i}\right)^{-1}\left(\frac{M_{\text{init}}}{m}+1\right)\right]\right).$$

Using the choice of $M_{\text{init}}$ and $M$, and the bound (16), we get the time complexity

$$= \tilde{\mathcal{O}} \left( \kappa \left( \frac{L_g}{\mu\varepsilon} + \frac{\sqrt{L_h}}{\mu^{3/4}\sqrt{\varepsilon}} \right) + \min_{m \in [n]} \left[ \left( \frac{1}{m} \sum_{i=1}^{m} \frac{1}{h_i} \right)^{-1} \left( \frac{L_g}{\mu\varepsilon} + \frac{\sqrt{L_h}}{\mu^{3/4}\sqrt{\varepsilon}} + \frac{\sigma^2}{m\mu\varepsilon^2} + \frac{\sigma^2\sqrt{L_h}}{m\mu^{7/4}\varepsilon^{5/2}} \right) \right] \right)$$

$$+ \tilde{\mathcal{O}} \left( \min_{m \in [n]} \left[ \left( \frac{1}{m} \sum_{i=1}^{m} \frac{1}{h_i} \right)^{-1} \left( \frac{\sigma^2}{m\mu\varepsilon^2} + 1 \right) \right] \right)$$

$$= \tilde{\mathcal{O}} \left( \kappa \left( \frac{L_g}{\mu\varepsilon} + \frac{\sqrt{L_h}}{\mu^{3/4}\sqrt{\varepsilon}} \right) + \min_{m \in [n]} \left[ \left( \frac{1}{m} \sum_{i=1}^{m} \frac{1}{h_i} \right)^{-1} \left( \frac{L_g}{\mu\varepsilon} + \frac{\sqrt{L_h}}{\mu^{3/4}\sqrt{\varepsilon}} + \frac{\sigma^2}{m\mu\varepsilon^2} + \frac{\sigma^2\sqrt{L_h}}{m\mu^{7/4}\varepsilon^{5/2}} \right) \right] \right).$$

It is left to substitute $h_i = \dot{h}_i \times H$ and recall that $H = \tilde{\mathcal{O}} \left( \frac{1}{1-\gamma} \right)$. $\qquad\square$

## F.2 TIME COMPLEXITY OF Malenia NIGT

**Theorem A.3.** *Consider the results and assumptions of Theorem A.1. Additionally, consider that Assumption 2.5 holds. The time required to find an $\varepsilon$–solution up to $\sqrt{2}\varepsilon'/\sqrt{\mu}$ neighborhood by Malenia NIGT (Algorithms 1 and 3) in the heterogeneous setup is*

$$\tilde{\mathcal{O}} \left( \kappa \left( \frac{L_g}{\mu\varepsilon} + \frac{\sqrt{L_h}}{\mu^{3/4}\sqrt{\varepsilon}} \right) + \frac{1}{1-\gamma} \left[ \dot{h}_n \left( \frac{L_g}{\mu\varepsilon} + \frac{\sqrt{L_h}}{\mu^{3/4}\sqrt{\varepsilon}} \right) + \left( \frac{1}{n} \sum_{i=1}^{n} \dot{h}_i \right) \left( \frac{\sigma^2}{n\mu\varepsilon^2} + \frac{\sigma^2\sqrt{L_h}}{n\mu^{7/4}\varepsilon^{5/2}} \right) \right] \right).$$

*Proof Sketch.* With our choice of $M$ and $M_{\text{init}}$, the number of global iterations is

$$T = \mathcal{O} \left( \left( \frac{L_g}{\mu\varepsilon} + \frac{\sqrt{L_h}}{\mu^{3/4}\sqrt{\varepsilon}} \right) \log \left( \frac{\Delta}{\varepsilon} \right) \right)$$

Notice that the time requires to collect $M$ stochastic gradients in Algorithm 3 is

$$\mathcal{O} \left( \kappa + h_n + \left( \frac{1}{n} \sum_{i=1}^{n} h_i \right) \frac{M}{n} \right).$$

due to Lemma C.2. Thus, after $T$ iterations, the total communication time is

$$\mathcal{O} \left( T \times \left[ \kappa + h_n + \left( \frac{1}{n} \sum_{i=1}^{n} h_i \right) \frac{M}{n} \right] \right) + \mathcal{O} \left( \kappa + h_n + \left( \frac{1}{n} \sum_{i=1}^{n} h_i \right) \frac{M_{\text{init}}}{n} \right)$$

$$= \mathcal{O} \left( \kappa T \right) + \mathcal{O} \left( h_n T \right) + \mathcal{O} \left( \left( \frac{1}{n} \sum_{i=1}^{n} h_i \right) \frac{MT}{n} \right) + \mathcal{O} \left( \kappa + h_n + \left( \frac{1}{n} \sum_{i=1}^{n} h_i \right) \frac{M_{\text{init}}}{n} \right).$$

Using the choice of $M_{\text{init}}$ and $M$, and the bound on $T$, the total communication time is

$$\mathcal{O} \left( \kappa T + h_n T + \left( \frac{1}{n} \sum_{i=1}^{n} h_i \right) \left( \frac{\sigma^2}{n\mu\varepsilon^2} + \frac{\sigma^2\sqrt{L_h}}{n\mu^{7/4}\varepsilon^{5/2}} \right) + \left( \frac{1}{n} \sum_{i=1}^{n} h_i \right) \frac{\sigma^2}{n\mu\varepsilon^2} \right)$$

$$\mathcal{O} \left( \kappa \left( \frac{L_g}{\mu\varepsilon} + \frac{\sqrt{L_h}}{\mu^{3/4}\sqrt{\varepsilon}} \right) + h_n \left( \frac{L_g}{\mu\varepsilon} + \frac{\sqrt{L_h}}{\mu^{3/4}\sqrt{\varepsilon}} \right) + \left( \frac{1}{n} \sum_{i=1}^{n} h_i \right) \left( \frac{\sigma^2}{n\mu\varepsilon^2} + \frac{\sigma^2\sqrt{L_h}}{n\mu^{7/4}\varepsilon^{5/2}} \right) \right).$$

It is left to substitute $h_i = \dot{h}_i \times H$ and recall that $H = \tilde{\mathcal{O}} \left( \frac{1}{1-\gamma} \right)$. $\qquad\square$

## G PROOF OF THE LOWER BOUND IN THE HOMOGENEOUS SETUP

In this section, we prove a lower bound for a family of methods that only access unbiased stochastic gradients of an $(L_g, L_h)$–twice smooth function $F$ with $\sigma$–bounded stochastic variance. We should clarify that our lower bound applies only to methods and proofs that use stochastic gradients (3) as a black-box oracle. In other words, our lower bound applies to all methods and proofs for which it

is sufficient to take Proposition 2.3 as an assumption with $L_g, L_h$, and $\sigma^2$ being constants. For our proof strategy, as well as the previous state-of-the-art strategies (Fatkhullin et al., 2023; Lan et al., 2025), this is the case. Extending the lower bound to methods that fully utilize the structure of $J$ in (1) is an important and challenging problem. Nevertheless, to the best of our knowledge, this is the first attempt to provide a lower bound for our asynchronous and distributed setting.

**Theorem G.1.** *For all $\Delta > 0$, $L_g, L_h, \varepsilon > 0$, and $\sigma > 0$ such that $\varepsilon \leq \mathcal{O}(\sigma)$, there exists an twice smooth function $F$ with $L_g$–Lipschitz gradients, $L_h$-Lipschitz Hessians, and $F(0) - F^* \leq \Delta$, and an oracle that returns unbiased stochastic gradients with $\sigma$–bounded gradient variance such that any first-order zero-respecting algorithm, where the agents communicate with the server or other agents to update an iterate, requires at least*

$$\tilde{\Omega} \left( \kappa \times \frac{L_1^{3/7} L_2^{2/7} \Delta}{\varepsilon^{12/7}} + H \min_{m \in [n]} \left[ \left( \frac{1}{m} \sum_{i=1}^{m} \frac{1}{h_i} \right)^{-1} \left( \frac{\sigma^2}{m\varepsilon^2} + 1 \right) \right] \times \min \left\{ \frac{L_g \Delta}{\varepsilon^2}, \frac{\sqrt{L_h} \Delta}{\varepsilon^{3/2}} \right\} \right).$$

*seconds under Assumption 2.5 to output an $\varepsilon$-stationary point with high probability.*

One particular direction is to extend the result to methods with variance-reduction techniques (Huang et al., 2020; Ding et al., 2022; Xu et al., 2020a;b; Fan et al., 2021), which rely on importance sampling (IS). Since the distribution of trajectories is non-stationary, such methods require an additional strong assumption that the IS weights are bounded. Our lower bound applies to all methods that do not require this additional assumption, and extending the lower bound to settings with these extra assumptions is an important direction for future work. Moreover, it would be interesting to extend the lower bound to Hessian-aided PG methods (Fatkhullin et al., 2023; Ganesh et al., 2024), which will also require taking into account an extra assumption in the design of lower bounds that the variance of Hessians is bounded.

## G.1    PROOF OF THEOREM G.1

*Proof.* We are slightly concise in descriptions since the proof almost repeats the ideas from (Arjevani et al., 2022; Carmon et al., 2021; Arjevani et al., 2020; Tyurin & Richtárik, 2023; 2024; Tyurin et al., 2024b). In particular, Tyurin & Richtárik (2023) provided the proof of the lower bound

$$\tilde{\Omega} \left( \min_{m \in [n]} \left[ \left( \frac{1}{m} \sum_{i=1}^{m} \frac{1}{h_i} \right)^{-1} \left( \frac{\sigma^2}{m\varepsilon^2} + 1 \right) \right] \times T \right)$$

for $L_g$–smooth functions $F$ without second-order smoothness, where $T = \Theta\left( \frac{L_g \Delta}{\varepsilon^2} \right)$. However, following (Arjevani et al., 2020) and using the same scaled "worst-case" function as in (Carmon et al., 2020; Arjevani et al., 2020; 2022), we have to take a different dimension

$$T = \Theta \left( \frac{\Delta}{\varepsilon} \min \left\{ \frac{L_g}{\varepsilon}, \frac{\sqrt{L_h}}{\sqrt{\varepsilon}} \right\} \right) = \Theta \left( \min \left\{ \frac{L_g \Delta}{\varepsilon^2}, \frac{\sqrt{L_h} \Delta}{\varepsilon^{3/2}} \right\} \right)$$

instead of $\frac{L_g \Delta}{\varepsilon^2}$ to ensure that the function has $L_h$–smooth Hessians (we take $T$ as in (84) from (Arjevani et al., 2020)). Thus, the lower bound is

$$\tilde{\Omega} \left( \min_{m \in [n]} \left[ \left( \frac{1}{m} \sum_{i=1}^{m} \frac{1}{h_i} \right)^{-1} \left( \frac{\sigma^2}{m\varepsilon^2} + 1 \right) \right] \times \min \left\{ \frac{L_g \Delta}{\varepsilon^2}, \frac{\sqrt{L_h} \Delta}{\varepsilon^{3/2}} \right\} \right).$$

It is left to substitute $h_i = \dot{h}_i \times H$. We can get the communication term using the deterministic construction from (Carmon et al., 2021), which says that required number call of the first-order oracle is

$$\Omega \left( \frac{L_1^{3/7} L_2^{2/7} \Delta}{\varepsilon^{12/7}} \right).$$

Thus, the lower bound for communication is $\Omega \left( \kappa \times \frac{L_1^{3/7} L_2^{2/7} \Delta}{\varepsilon^{12/7}} \right)$ seconds under the assumption that the agents communicate with the server or other agents after every gradient computation.    □

# H    EXPERIMENTS

In our experiments, we compare the performance of Rennala NIGT with AFedPG by Lan et al. (2025) and with the synchronized version of NIGT (Synchronized NIGT), where all agents compute one stochastic gradient, aggregate them in a synchronized fashion, and perform the standard NIGT step (Fatkhullin et al., 2023). We focus on the MuJoCo tasks (Todorov et al., 2012). We consider the standard setup, where the actions are sample from a Gaussian policy. Given a state $s$, the policy outputs mean $\mu_\theta(s)$ and standard deviation $\sigma_\theta(s) > 0$, and samples

$$u_t \sim \pi_\theta(\cdot \mid s) = \mathcal{N}\big(\mu_\theta(s), \mathrm{diag}(\sigma_\theta^2(s))\big).$$

Then, the actions are defined as $a_t = \alpha \tanh(u_t)$, where $\alpha$ is an appropriate scaling factor (for the most MuJoCo tasks, $\alpha = 1$). We take the neural network architecture $s \to \mathrm{Linear}(d_s, 64) \to \mathrm{Tanh} \to \mathrm{Linear}(64, 64) \to \mathrm{Tanh} \to \{\mathrm{Linear}(64, d_a) \to \mu_\theta, \mathrm{Linear}(64, d_a) \to \mathrm{Softplus} \to \sigma_\theta\}$, where $d_s$ and $d_a$ are the dimensions of the state and action spaces, respectively.

We consider the centralized setting with different computation and communication scenarios, with $h_i$ denoting the computation time of agent $i$ and $\kappa_i$ denoting the communication time for sending one vector from agent $i$ to the server. For instance, if $h_i = 1$ and $\kappa_i = 0$, then agent $i$ computes one gradient in 1 second and sends it to the server without delay. Unlike Section 2.3, in the experimental part we consider a more general setting where agents have different communication times. However, Assumption 2.5 still holds with $\kappa = \max\limits_{i \in [n]} \kappa_i$.

We run every experiment with 5 seeds and report $(20\%, 80\%)$ confidence intervals. All methods start from the same point, for a fixed seed, and have two parameters: the momentum $\eta \in \{0.001, 0.01, 0.1\}$ and the learning rate $\alpha \in \{2^{-10}, 2^{-9}, \ldots, 2^{-1}\}$. We tune both on the Humanoid-v4 task with equal computation speeds and zero communication delays and observe that $\eta = 0.1$ is the best choice for all algorithms. However, $\alpha$ is tuned differently for different algorithms. Thus, in the following experiments, we tune $\alpha \in \{2^{-10}, 2^{-9}, \ldots, 2^{-1}\}$ for every plot and algorithm. Our algorithm Rennala NIGT has the additional parameters $M$ and $M_{\mathrm{init}}$. We take $M_{\mathrm{init}} = M$ which is tuned as $M \in \{20, 30, 50\}$.

The code was written in Python 3 using PyTorch (Paszke et al., 2019). The distributed environment was emulated on machines with Intel(R) Xeon(R) Gold 6278C CPU @ 2.60GHz and 52 CPUs.

## H.1    EXPERIMENTS WITH DIFFERENT ENVIRONMENTS

We start with the experiment on Humanoid-v4 from the main part of the paper (see Figure 1 or Figure 2). We consider horizon 512 and $n = 10$ agents with equal computation speeds and zero communication delays: $h_i = 1$ and $\kappa_i = 0$ for all $i \in [n]$, and observe that the performance of all methods is almost the same, which is expected. Then, we increase the heterogeneity of times by taking $h_i = \sqrt{i}$ and $\kappa_i = \sqrt{i}$ for all $i \in [n]$, and $h_i = \sqrt{i}$ and $\kappa_i = \sqrt{i} \times d^{1/4}$ for all $i \in [n]$, where $d$ is the number of parameters. We observe that Rennala NIGT is the only robust method, and converges faster than other methods.

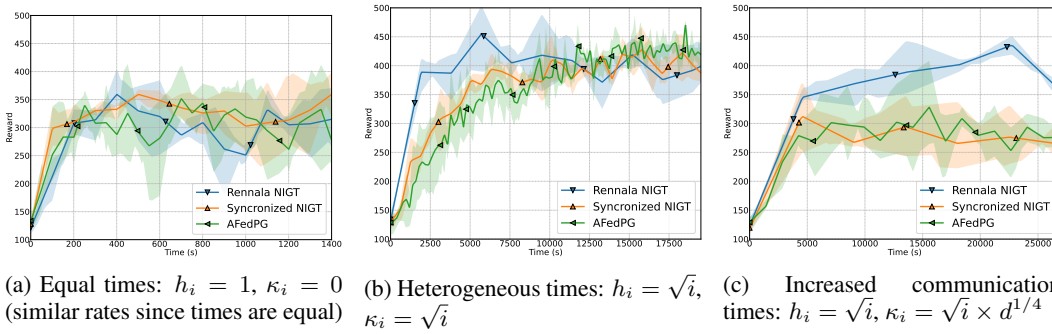

(a) Equal times: $h_i = 1$, $\kappa_i = 0$ (similar rates since times are equal)

(b) Heterogeneous times: $h_i = \sqrt{i}$, $\kappa_i = \sqrt{i}$

(c)    Increased    communication times: $h_i = \sqrt{i}$, $\kappa_i = \sqrt{i} \times d^{1/4}$

Figure 2: Experiments on Humanoid-v4 with increasing heterogeneity of times (from left to right).

We also consider other environments in Figure 3: Reacher-v4 with horizon $1024$, Walker2d-v4 with horizon $1024$, and Hopper-v4 with horizon $1024$. We observe that our algorithm converges faster on Humanoid-v4 and Reacher-v4. However, for Walker2d-v4 and Hopper-v4, the gap between the algorithms is less pronounced.

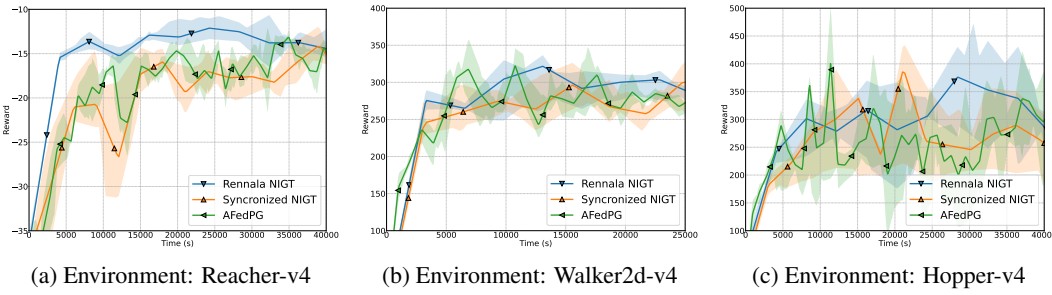

| (a) Environment: Reacher-v4 | (b) Environment: Walker2d-v4 | (c) Environment: Hopper-v4 |

Figure 3: Experiments on MuJoCo tasks with $h_i = \sqrt{i}$, $\kappa_i = \sqrt{i} \times d^{1/4}$.

## H.2 INCREASING NUMBER OF AGENTS AND MORE SCENARIOS

We now verify whether Rennala NIGT performs better as the number of agents increases to $n = 100$. Moreover, we examine different computation scenarios to validate the robustness of Rennala NIGT. Once again, in the equal-times case shown in Figure 4, all algorithms scale with the number of agents and exhibit similar performance, which is expected.

Next, we consider four heterogeneous computation and communication scenarios: i) In Figure 5, communication is free while computation times are heterogeneous. We observe that Rennala NIGT converges faster; ii) In Figure 6, communication times are equal to computation times. Here, Rennala NIGT still achieves the best convergence rate; iii) In Figure 7, we increase the communication time and observe that the performance gap also increases: Rennala NIGT is significantly faster, which aligns with our theoretical results (see Table 1); iv) Finally, in Figure 8, we examine the case where computation times decrease and find that Rennala NIGT remains the fastest method.

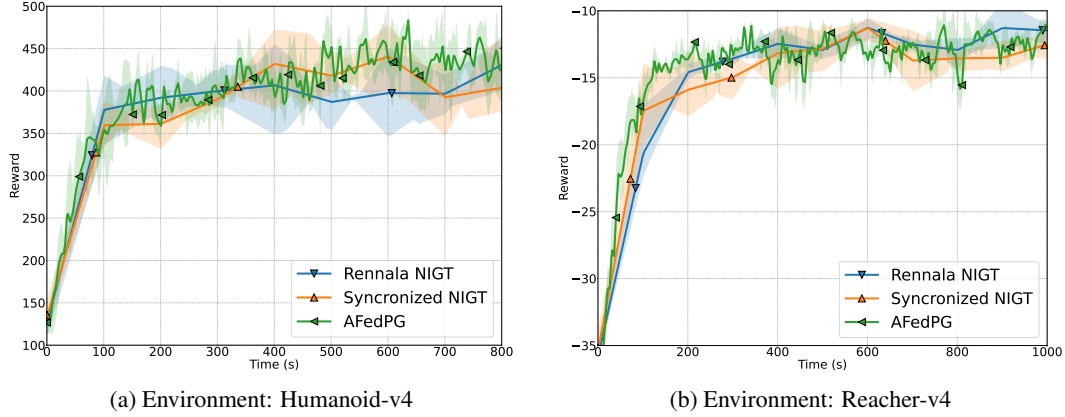

| (a) Environment: Humanoid-v4 | (b) Environment: Reacher-v4 |

Figure 4: Experiments on MuJoCo tasks with $h_i = 1$, $\kappa_i = 0$ and $n = 100$ (similar rates since times are equal).

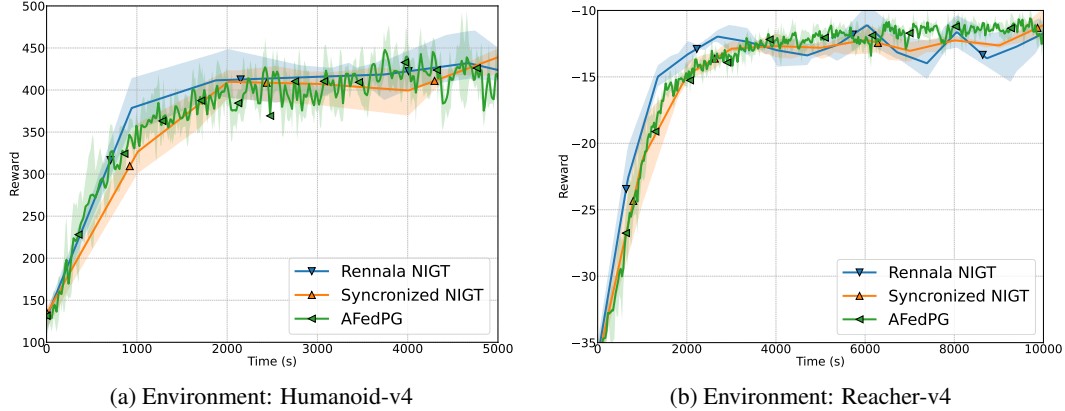

(a) Environment: Humanoid-v4        (b) Environment: Reacher-v4

Figure 5: Experiments on MuJoCo tasks with $h_i = \sqrt{i}$, $\kappa_i = 0$ and $n = 100$.

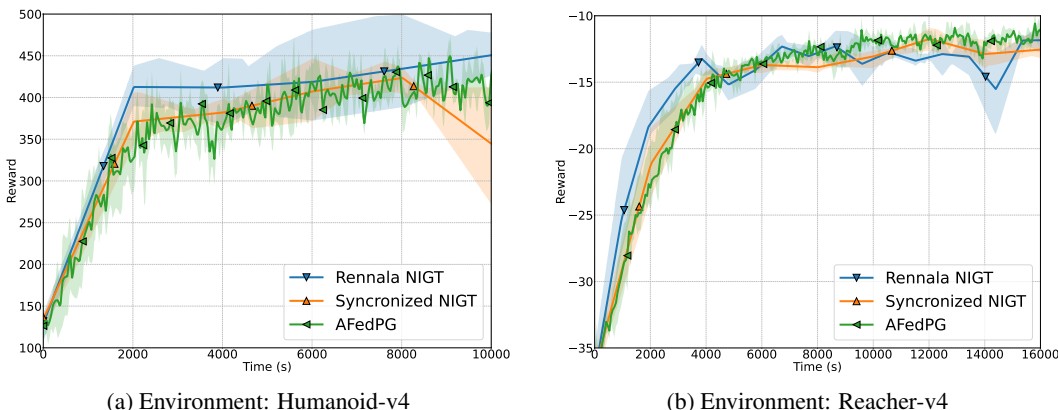

(a) Environment: Humanoid-v4        (b) Environment: Reacher-v4

Figure 6: Experiments on MuJoCo tasks with $h_i = \sqrt{i}$, $\kappa_i = \sqrt{i}$ and $n = 100$.

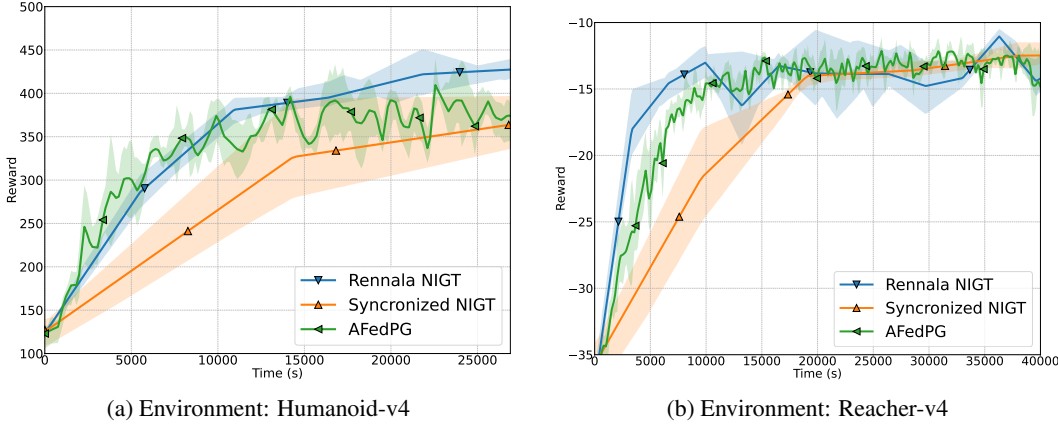

(a) Environment: Humanoid-v4        (b) Environment: Reacher-v4

Figure 7: Experiments on MuJoCo tasks with $h_i = \sqrt{i}$, $\kappa_i = \sqrt{i} \times d^{1/4}$ and $n = 100$.

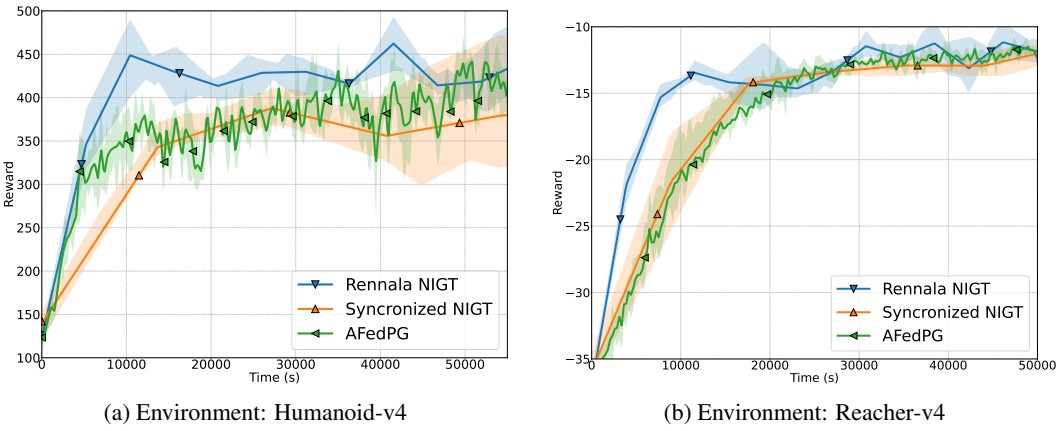

(a) Environment: Humanoid-v4          (b) Environment: Reacher-v4

Figure 8: Experiments on MuJoCo tasks with $h_i = i^{1/4}$, $\kappa_i = \sqrt{i} \times d^{1/4}$ and $n = 100$.

## H.3 HETEROGENEOUS SETTING

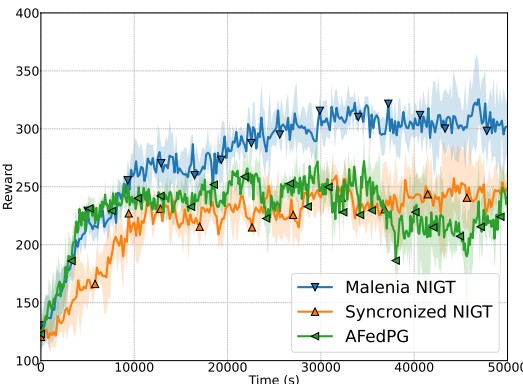

Figure 9: Experiments on Humanoid-v4 in the heterogeneous setting.

In this section, we compare Malenia NIGT, a method designed for heterogeneous setups. This section repeats the previous experimental setup with one important difference: the agents sample trajectories from different environments. We take two agents, $n = 2$. The first agent has access to the standard Humanoid-v4 environment from MuJoCo. However, the second agent receives *inverted* states. More formally, when the first worker performs action $a_t$, the Humanoid-v4 environment returns state $s_{t+1}$, and instead of immediately receiving it, we concatenate the value 0 to $s_{t+1}$, and finally redirect $(s_{t+1}, 0)$ to the first agent. In the case of the second worker, we redirect $(-s_{t+1}, 1)$, where we multiply the state by $-1$ to invert it. Thus, the second worker gets inverted states. We concatenate the values 0 and 1 to indicate the type of environment to the agents and to the model. This way, we can compare algorithms and analyze their capability to handle heterogeneous environments. We present the results for $n = 2$ workers with $h_0 = 1$, $h_1 = 10$, and no communication overhead, emulating the scenario where one worker is much faster.

In Figure 9, we can see a large gap between Malenia NIGT and AFedPG: Malenia NIGT achieves a much higher reward, as expected, since it provably supports both heterogeneous computations and heterogeneous environments (see Section 5 and Table 2).

