# OpenReview forum: "Asynchronous Policy Gradient Aggregation for Efficient Distributed Reinforcement Learning"
_ICLR.cc/2026/Conference — ICLR 2026 Poster_

### Official Review · Reviewer_KcX2 · 2025-10-16

**Soundness:** 4
**Presentation:** 4
**Contribution:** 4
**Rating:** 8
**Confidence:** 4

**Summary:**

This paper studies distributed reinforcement learning (RL) with asynchronous policy gradient methods. The authors propose two new algorithms: Rennala NIGT for homogeneous environments and Malenia NIGT for heterogeneous environments. Both methods extend normalized policy gradient techniques with asynchronous aggregation, achieving improved computational and communication complexity compared to prior work, e.g., AFedPG. The authors also establish a new lower bound and validate the methods on MuJoCo tasks, showing robustness to heterogeneity and communication delays.

**Strengths:**

**Originality.** Proposes new algorithms that generalize normalized gradient techniques to asynchronous distributed RL, with contributions in both homogeneous and heterogeneous settings.

**Quality.** Strong theoretical analysis, including new upper bounds, support for AllReduce, and a new lower bound. Experimental results align with the theory.

**Clarity.** The paper is well-structured, with clear algorithm pseudocode and comparisons.

**Significance.** Distributed RL is critical for scaling up to large systems. The improved communication and computation guarantees, especially with AllReduce support, are practically valuable.

**Weaknesses:**

**Optimality gap** There remains a gap between the presented upper bounds (κε⁻²) and the lower bound (κε⁻¹²/⁷), leaving open whether the methods are near-optimal.

**Questions:**

In Malenia NIGT, the mean-like dependency on agent speeds is less favorable than the harmonic dependency in Rennala NIGT. How does this affect performance in highly heterogeneous environments?

---

> ### Author Response · Authors · 2025-11-14
>
> Thank you for your positive review!
>
> > There remains a gap between the presented upper bounds (κε⁻²) and the lower bound (κε⁻¹²/⁷), leaving open whether the methods are near-optimal.
>
> Yes, this is true. We obtain the latter term by reducing our problem to the lower bound of [1] for $(L_g, L_h)$–twice smooth functions. To the best of our knowledge, it remains an open problem whether the $\varepsilon^{-12/7}$ rate can be achieved, even in the non-distributed deterministic case. Thus, it appears to be a very difficult problem.
>
> > In Malenia NIGT, the mean-like dependency on agent speeds is less favorable than the harmonic dependency in Rennala NIGT. How does this affect performance in highly heterogeneous environments?
>
> Let us explain this part. Ignoring the communication terms and assuming small $\varepsilon$, for Rennala NIGT we obtain the harmonic-like complexity
>
> $$(R) = {\min\limits_{m \in [n]}\left[\left(\frac{1}{m}\sum\limits^m_{i=1}\frac{1}{\dot{h}_i}\right)^{-1}\left(\frac{1}{\varepsilon^2} + \frac{1}{m \varepsilon^{7/2}}\right)\right]},$$
>
> while for Malenia NIGT we get the mean-like complexity
>
> $(M) = \left(\frac{1}{n}\sum\limits^n_{i=1} \dot{h}_i\right) \frac{1}{n \varepsilon^{7/2}}.$
>
> If $\dot{h}_n \to \infty$, then the complexity in the homogeneous setting remains stable and one can show that
>
> $$(R) = {\min\limits_{m \in [n - 1]}\left[\left(\frac{1}{m}\sum\limits^m_{i=1}\frac{1}{\dot{h}_i}\right)^{-1}\left(\frac{1}{\varepsilon^2} + \frac{1}{m \varepsilon^{7/2}}\right)\right]},$$
>
> where we take $\min\limits_{m \in [n - 1]}$ instead of $\min\limits_{m \in [n]}.$ The method and formula automatically ignore the slowest worker and continue working as if the slowest worker does not exist.
>
> However, in the heterogeneous setting, the time complexity tends to $\infty$ due to the linear dependency on ${\dot{h}_i}$ in $(M).$ Thus, if one worker is very slow, then the ratio $(M) / (R)$ can be arbitrarily large.
>
> In general, there are many regimes where $(M)/(R) \gg 1$ because the mean of the values is never smaller than their harmonic mean, while the mean can be arbitrarily larger.
>
> Thank you! If you have more questions, please let us know.
>
> ---
>
> [1]: Yair Carmon, John C Duchi, Oliver Hinder, and Aaron Sidford. Lower bounds for finding stationary points ii: first-order methods. Mathematical Programming, 185(1):315–355, 2021.

---

### Official Review · Reviewer_GmQ8 · 2025-11-02

**Soundness:** 3
**Presentation:** 3
**Contribution:** 3
**Rating:** 6
**Confidence:** 3

**Summary:**

This paper investigates distributed policy gradient (PG) methods in asynchronous and parallel reinforcement learning (RL) settings. The authors introduce two new algorithms for homogeneous environments and heterogeneous environments. Both algorithms are based on normalized policy gradient methods and asynchronous aggregation schemes inspired by recent work (e.g. Lan et al. 2025)

**Strengths:**

* I briefly checked the proofs and they seem correct. This means the paper improves over the communication complexity of existing methods.

* They also show a lower bound for the homogenous setting in the appendix which seems to show there is room for improvement.

**Weaknesses:**

* There is a lack of novelty in the algorithmic components, while one may argue this particular combination is done for the first time.

* On the theoretical side, the proofs are highly reliant on/similar to existing results in federated RL and stochastic optimization. For instance, Lemma D.2 is a standard result (similar to SGDm with normalization and also a corresponding lemma from Lan et al. 2025). What is a new technical result proposed by the paper?

* I also have a concern about the heterogenous experiments in the Appendix H.3. To test Malenia NIGT in a scenario with environment heterogeneity (different $J_i$), the paper uses $n=2$ agents, where one agent receives state $s$ and the other receives state $-s$. To distinguish these, the paper states, "we concatenate the value 0 to $s_{t+1}$... In the case of the second worker, we redirect $(-s_{t+1}, 1)$". In my view, however, this is not a heterogeneous problem. The algorithm is not learning an average policy $\frac{1}{2}(J_1(\theta) + J_2(\theta))$. It is learning a single policy $J'(\theta)$ for a single environment whose state space is $\mathcal{S} \times \{0, 1\}$. The policy network can trivially learn to behave differently based on the appended bit.

**Questions:**

Please see the Weakness section.

---

> ### Author Response · Authors · 2025-11-14
>
> Thank you for your review!
>
> > There is a lack of novelty in the algorithmic components, while one may argue this particular combination is done for the first time.
>
> We tried to understand the limits of improving the previous state-of-the-art method by Lan et al. (2025) in the distributed RL setting. Through this work, we established a new lower bound in the homogeneous setting and identified a gap between this lower bound and the result of Lan et al. (2025), in addition to the fact that their method does not support heterogeneous environments. This led us to ask whether we could develop a new method that both improves the previous complexity and bridges this gap. We designed two such methods: Rennala NIGT and Malenia NIGT. Without Rennala NIGT, the gap would be substantially larger, which highlights the necessity and contribution of our work.
>
> Showing that this gap can be improved with simpler methods and proofs is an advantage rather than a disadvantage. It would also be interesting for RL researchers and engineers that it is possible to obtain improvements using our asynchronous aggregation techniques, which can be applied in other RL contexts. Notice that [1] worked with the classical SGD method. We worked with the NIGT method, which has much more complex steps. While it is now clear that our approach works, this was not fully clear to us beforehand.
>
> > On the theoretical side, the proofs are highly reliant on/similar to existing results in federated RL and stochastic optimization.
>
> We did not naively reuse the analysis, as one might think. For example, the analysis in [2] is not tight. It is true that they have the correct dependence on $T$ in their rate in Theorem 5: $O\left(\frac{1}{(T + 1)^{2/7}}\right)$. However, the dependence on the noise $L_h$ is suboptimal. Due to their last term $\frac{(L_h + \sigma_g)\log(T + 1)}{(T + 1)^{2/7}}$ in Theorem 5 of the appendix, their complexity is $\frac{(L_h + \sigma_g)}{\varepsilon^{7/2}}$, which is suboptimal when $\sigma$ is small or equal to zero. Indeed, taking $\sigma_g = 0$, their rate becomes $\frac{(L_h + \sigma_g)}{\varepsilon^{7/2}}$, while our rate is $\frac{L_g}{\varepsilon^{2}}$ in Theorem 4.1. We have a much better dependence on $\varepsilon$ when $\sigma=0.$ This improvement was crucial for the entire theory to work because our asynchronous aggregation techniques effectively reduce the influence of $\sigma$. There are several other similar improvements in our analysis. Additionally, we proved the lower bound to show that Rennala NIGT is almost optimal, which we believe is an important contribution to the community.
>
> > I also have a concern about the heterogenous experiments in the Appendix H.3.
>
> This is a heterogeneous setting because, mathematically, the trajectories $(s_{t}, 0)$ and $(-s_{t}, 1)$ come from different distributions (e.g., just as the normal distributions $\mathcal{N}(-1, 0)$ and $\mathcal{N}(1, 0)$ are different). For us as humans, we can easily reduce this problem to a homogeneous one using the last bit, but for a neural network model, this is a truly heterogeneous setting: it has to learn this relationship. This toy example allows us to analyze the ability of algorithms to handle heterogeneous environments and how quickly the algorithms can learn and adapt to different distributions. In more realistic heterogeneous scenarios, the neural network might also have to find a way to distinguish the distributions by learning the differences between them.
>
> **Thank you for the comments! If you have more questions, please let us know.**
>
> ---
>
> [1]: Alexander Tyurin and Peter Richtarik. Optimal time complexities of parallel stochastic optimization methods under a fixed computation model. Advances in Neural Information Processing Systems, 2023.
>
> [2]: Ilyas Fatkhullin, Anas Barakat, Anastasia Kireeva, and Niao He. Stochastic policy gradient methods: Improved sample complexity for fisher-non-degenerate policies. In International Conference on Machine Learning, pp. 9827–9869. PMLR, 2023.

---

### Official Review · Reviewer_Codc · 2025-11-07

**Soundness:** 3
**Presentation:** 2
**Contribution:** 2
**Rating:** 4
**Confidence:** 4

**Summary:**

This paper introduces two asynchronous distributed policy-gradient algorithms for reinforcement learning:

* Rennala NIGT for the *homogeneous* setting (all agents share the same environment), and
* Malenia NIGT for the *heterogeneous* setting (each agent may have a different environment).

Both algorithms combine the NIGT policy gradient update (Fatkhullin et al., 2023) with asynchronous aggregation mechanisms adapted from recent federated SGD literature (Tyurin & Richtárik, 2023). The paper provides wall-clock convergence guarantees (in terms of computation and communication time) to reach an ε-stationary point under standard policy gradient assumptions. A lower bound for twice-smooth stochastic objectives is also presented, and empirical results on MuJoCo tasks illustrate the wall-clock speedups.

**Strengths:**

* The problem setting (efficient RL under asynchronous environments even with heterogeneity) is important and highly relevant.
* The paper is clearly written and the proofs are easy to follow.

**Weaknesses:**

**Novelty.** Conceptually, I struggle to see what is new here. The algorithm appears to essentially be a direct combination of two known components:

* Rennala/Malenia from federated SGD (Tyurin & Richtárik, 2023), and
* NIGT-based policy gradient from non-distributed RL (Fatkhullin et al., 2023).

The analysis likewise seems to be a straightforward combination. In particular, the core convergence argument (Section D, also F) closely follows the standard NIGT analysis, and relies on the fact that Rennala/Malenia provide unbiased policy gradient estimates with bounded variance (∝ 1/M). While it is normal for new work to build on prior components, here it is unclear where the main technical challenge lies (if any).

**Lower bound and Missing Comparisons.** The paper’s main theoretical focus is ε-stationary convergence and also provide a lower bound for this rate (which is assuming access to only stochastic gradient estimates and no other information). The setting for the lower bound feels very artificial especially since there is a substantial body of RL papers that obtain improved ε-stationary rates by exploiting RL-specific structure (see [A],[B],[C] and also Hessian aided PG in (Fatkhullin et al 2023)). It is very odd that these are not even mentioned, let alone discussed or compared against.


Finally, although this is less critical, the paper should be better contextualized relative to existing federated PG work. Apart from (Lan et al. 2025), the empirical comparisons focus on non-federated PG methods, which are not the most natural baselines here. There are federated PG methods with ε-stationary guarantees in the synchronous homogeneous case [C], federated PG based on Hessian aided PG in the same regime [D], and heterogeneous PG methods using softmax policies [E]. Including (or at least discussing) such baselines would better position this work within the federated PG literature.



---
[A] Xu, Gao, Gu. Sample Efficient Policy Gradient Methods with Recursive Variance Reduction. ICLR 2020.

[B] Xu, Gao, Gu. An Improved Convergence Analysis of Stochastic Variance–Reduced Policy Gradient. UAI 2020.

[C] Fan, Ma, Dai, Jing, Tan. Fault–Tolerant Federated Reinforcement Learning with Theoretical Guarantee. NeurIPS 2021.

[D] Ganesh, Chen, Thoppe, Aggarwal. Global Convergence Guarantees for Federated Policy Gradient Methods with Adversaries. TMLR 2024.

[E] Labbi, Mangold, Tiapkin, Moulines. On Global Convergence Rates for Federated Policy Gradient under Heterogeneous Environment. arXiv 2025.

**Questions:**

Please see weaknesses listed.

---

> ### Author Response · Authors · 2025-11-14
> **Official Comment by Authors (Part 1)**
>
> Thank you for your review. We now respond to the comment.
>
> > Conceptually, I struggle to see what is new here. …
>
> Our goal was to understand the limits and possibilities for improving the previous state-of-the-art method by Lan et al. (2025) in the timely distributed RL setting. We discovered a new lower bound in the homogeneous setting and identified a gap between this lower bound and the result of Lan et al. (2025), in addition to the fact that their method does not support heterogeneous environments. A natural next step was therefore to develop a new method that could bridge this gap. In this work, we introduce two such methods, Rennala NIGT and Malenia NIGT. Without developing the Rennala NIGT method, the gap would be much larger, which further justifies the necessity and contribution of our work.
>
> We believe that demonstrating that this gap can be reduced using simpler methods and proofs is a strength of our paper rather than a weakness. It is also interesting for the RL community that this improvement can be achieved using asynchronous aggregation techniques, which can be applied in other methods RL methods. Notice that [F] (Tyurin & Richtárik, 2023) were analyzing the vanilla SGD method with simple steps $x^{k+1} = x^k - \gamma \nabla f(x^k),$ while we were analyzing the NIGT method with much more complex steps. Knowing in advance that our approach would work wasn't clear.
>
> While it may appear that we simply copied and pasted the analysis, we did not. For instance, the analysis by [G] is not tight enough, because while they have the correct dependence on $T$ in their rate of Theorem 5 ($O(\frac{1}{(T + 1)^{2/7}})$), the dependence on the noise $L_h$ is suboptimal: look at the last term $\frac{(L_h + \sigma_g) \log(T + 1)}{(T + 1)^{2/7}}$ of their Theorem 5 in the appendix. Due to this term, their complexity to find an $\varepsilon$-stationary point is $\frac{(L_h + \sigma_g)}{\varepsilon^{7/2}}$
> To understand the difference, take $\sigma_g = 0,$ then their complexity would be $\frac{(L_h + \sigma_g)}{\varepsilon^{7/2}},$ while our iteration complexity in Theorem 4.1 is $\frac{L_g}{\varepsilon^{2}}$ with much better dependence on $\varepsilon.$ This improvement was essential for making the entire approach work and for obtaining an almost optimal time-complexity result, because our asynchronous aggregation approach precisely reduces the influence of $\sigma$, and we would get stuck with the wrong complexity without improving the analysis. Several other improvements were also required.

---

> > ### Author Response · Authors · 2025-11-14
> > **Official Comment by Authors (Part 2)**
> >
> > > The paper’s main theoretical focus is ε-stationary convergence and also provide a lower bound for this rate (which is assuming access to only stochastic gradient estimates and no other information).
> >
> > Thank you for the relevant references, which we have already added to the paper. Notice that [A, B, C] use the standard variance-reduction technique, rely on importance sampling (IS), and require an additional strong assumption (e.g., Assumption 4.4 in [A]) that the IS weights are bounded, which our approach avoids. In the RL setting, a sampled trajectory $\tau$ depends on the current policy $\theta$ (the distribution of $\tau$ is non-stationary). These methods, using variance-reduction techniques, evaluate stochastic policy gradients at different points but use the same trajectory $\tau,$ which leads to one of the stochastic policy gradients being biased. Introducing the importance-sampling weight and the additional strong assumption resolves this for them; however, our approach avoids these overcomplications and additional assumptions. Our lower bound applies to all methods that do not require the additional assumption, and extending the lower bound to settings with these extra assumptions is an important direction for future work.
> >
> > Investigating Hessian-aided PG is also an important direction; however, in practice, “the stochastic Hessian–vector products sampled in (N)-HARPG have much higher variance than stochastic gradients” (quote from [G]), which is why we focused on first-order methods.
> >
> > > Finally, although this is less critical, the paper should be better contextualized relative to existing federated PG work....
> >
> > Notice that we include the most natural baseline, Synchronized NIGT, a trivial extension of the classical NIGT method where all agents synchronize after every iteration. The discussion of other methods has already been included in the revision. We have extended our related work section and extended the review.
> >
> > **We believe that we have addressed all the comments and questions. Please take a look at the updated PDF, where we have incorporated your reference recommendations. If you have any further questions, please let us know.**
> >
> > ---
> >
> > [A] Xu, Gao, Gu. Sample Efficient Policy Gradient Methods with Recursive Variance Reduction. ICLR 2020.
> >
> > [B] Xu, Gao, Gu. An Improved Convergence Analysis of Stochastic Variance–Reduced Policy Gradient. UAI 2020.
> >
> > [C] Fan, Ma, Dai, Jing, Tan. Fault–Tolerant Federated Reinforcement Learning with Theoretical Guarantee. NeurIPS 2021.
> >
> > [D] Ganesh, Chen, Thoppe, Aggarwal. Global Convergence Guarantees for Federated Policy Gradient Methods with Adversaries. TMLR 2024.
> >
> > [E] Labbi, Mangold, Tiapkin, Moulines. On Global Convergence Rates for Federated Policy Gradient under Heterogeneous Environment. arXiv 2025.
> >
> > [F]: Alexander Tyurin and Peter Richtarik. Optimal time complexities of parallel stochastic optimization methods under a fixed computation model. Advances in Neural Information Processing Systems, 2023.
> >
> > [G]: Ilyas Fatkhullin, Anas Barakat, Anastasia Kireeva, and Niao He. Stochastic policy gradient methods: Improved sample complexity for fisher-non-degenerate policies. In International Conference on Machine Learning, pp. 9827–9869. PMLR, 2023.

---

> > > ### Author Response · Authors · 2025-11-27
> > >
> > > Dear Reviewer,
> > >
> > > Thank you for your time and effort! In this comment, we are kindly asking whether any of our answers are still unclear. We will be happy to respond to any questions. We hope that we have clarified our contributions and explained the differences between our approach and assumptions and those of the other papers you mentioned. All clarifications, missing discussion, and references are already in the updated PDF.
> > >
> > > Thank you!
> > >
> > > Authors

---

> ### Comment · Reviewer_Codc · 2025-11-28
> **Response by Reviewer Codc**
>
> I thank the authors for their detailed response and apologize for the delay in replying. I appreciate the clarification regarding the lower bound and the comparison with related work. I have reviewed the updated paper, and I would also encourage the authors to include a brief discussion on the methods that achieve improved stationary convergence rates via importance sampling or Hessian-aided methods in their section with the lower bound, and to clarify why such approaches are not considered here (similar to what is provided in the rebuttal).
>
> Regarding the novelty, while the discussion in the rebuttal is helpful, I still remain somewhat unconvinced. The rebuttal notes that “several other improvements were also required” - could the authors elaborate on these contributions? Overall, my only concern is novelty relative to prior work, and therefore my score remains unchanged. In its current form, I view the paper as slightly below the acceptance threshold, though I would not mind if it is accepted.

---

> ### Author Response · Authors · 2025-11-28
>
> Thank you for the response!
>
> > [...] I have reviewed the updated paper, and I would also encourage the authors to include a brief discussion on the methods that achieve improved stationary convergence rates via importance sampling or Hessian-aided methods in their section with the lower bound, and to clarify why such approaches are not considered here (similar to what is provided in the rebuttal).
>
> We've added this important discussion to Section G in the updated PDF.
>
> > Regarding the novelty, while the discussion in the rebuttal is helpful, I still remain somewhat unconvinced. The rebuttal notes that “several other improvements were also required” - could the authors elaborate on these contributions?
>
> As we explained previously, we have improved the dependence on $L_h, L_g,$$\sigma,$ and $\varepsilon$ to obtain a much tighter iteration complexity result in Theorem 4.1. Another improvement compared to [1] is that we consider the realistic scenario where the horizon $H$ is finite. Next, our work is the first to incorporate the time-complexity tools (Lemma C.1 and Lemma C.2) into Theorem 4.1 and to obtain the completely new Theorems 4.2, 4.4, 5.1, A.2, and A.3 with improved time complexities.
>
> Notice that in order to obtain Theorems 4.2, 4.4, 5.1, A.2, and A.3, we must choose particular values of $M$ and $M_{\text{init}}$, and we do not have much freedom here. In our initial research attempts, before improving the iteration analysis of previous works, we could not find values of $M$ and $M_{\text{init}}$ that would yield the time-complexity results in Theorems 4.2, 4.4, 5.1, A.2, and A.3. The reason was precisely the gap in the previous iteration rates, which we have now fixed. Observing the whole picture, improving it, and taking care of the details is a non-trivial and important effort in obtaining the final result.
>
> We want to emphasize that our contribution should be measured by the final computational and communication time-complexity results, which are significantly better than those of previous approaches and are nearly optimal (see Tables 1 and 2).
>
> > Overall, my only concern is novelty relative to prior work, and therefore my score remains unchanged. In its current form, I view the paper as slightly below the acceptance threshold, though I would not mind if it is accepted.
>
> We want to reiterate and explain that the novelty of our method is substantial. Compared to the previous AFedPG approach [1], (i) our *communication time complexity is provably better* in the homogeneous setup: Rennala NIGT significantly improves AFedPG; (ii) our *computational time complexity is also strictly better* in the homogeneous setup; and (iii) we developed a new method that *supports the heterogeneous setup* (unlike [1]). These improvements are obtained within the same setting and without requiring additional assumptions, and they are very close to the *lower bound that we also developed*.
>
> **Tables 1 and 2 clearly demonstrate the improvements and novelties compared to the previous results: we obtain much better time complexities, and Rennala NIGT achieves a time complexity very close to our developed lower bound.**
>
> Thank you for your time and effort!
>
> ---
>
> [1]: G. Lan, D. Han, A. Hashemi, V. Aggarwal, and C. G Brinton. Asynchronous federated reinforcement learning with policy gradient updates: Algorithm design and convergence analysis. (ICLR 2025)
>
> [2]: Ilyas Fatkhullin, Anas Barakat, Anastasia Kireeva, and Niao He. Stochastic policy gradient methods: Improved sample complexity for fisher-non-degenerate policies. (ICML 2023)

---

### Author Response · Authors · 2025-12-02
**Public Summary Comment by Authors**

Dear Reviewers, AC, SAC, and PCs,

We appreciate your time and effort in handling our submission. In this message, we want to summarize the submission and review process.

**(Our contributions):** In this paper, we developed two new methods, Rennala NIGT and Malenia NIGT, designed to provide new theoretical and practical state-of-the-art results in parallel and asynchronous settings. Our methods provably improve all previous methods, including our main competitor, the AFedPG method [1], under the same setting and without requiring additional assumptions (see Tables 1 and 2 in the paper).

Reviewers *GmQ8* and *KcX2* are positive about our work and recommend acceptance. **Reviewers *GmQ8* and *KcX2* agree in their comments that our communication and computational time complexities are provably better than those in previous approaches. Moreover, the reviewers also agree that we developed the lower bound in the homogeneous setting, allowing us to show that Rennala NIGT is much closer to the lower bound than other methods.**

Reviewer *Codc* outlined two weaknesses: "Novelty" and "Lower Bound and Missing Comparisons":
- The "Lower Bound and Missing Comparisons" weakness is already fixed and incorporated in the updated paper PDF (see the blue text around lines 060, 094, and 1300): all of the reviewer’s suggestions and missing references are discussed in the paper.
- The "Novelty" weakness states that we don't provide anything new. We respectfully disagree with this statement. In **(Our contributions)** we explain that our work provides the new state-of-the-art results that are clearly novel and significantly stronger than previous ones, which is also outlined by Reviewers *GmQ8* and *KcX2* in their comments. The novelty is evident and transparent in Tables 1 and 2. We clearly obtain a much better dependence on $\varepsilon,$ $n,$ and $\\{h_i\\}_i$ without requiring any additional assumption. Moreover, the significance of contributions should be measured by the improvements in complexities, not by the fact that the methods and proofs are “straightforward.” The fact that our results are substantially better while the proofs and methods remain concise and easy to follow is, in our view, an additional strength of the work. Finally, while it may appear to the reviewer that the methods and proofs are simple results, we want to emphasize that our path to the final result was not as simple as one might think. In the comments to the reviewer, we explain that we did not simply copy and paste the previous results and approaches: (i) the previous analysis was not tight enough and required additional modifications, and (ii) while it is now clear that our approach is better, at the beginning of our research it was not at all obvious that the proofs, the choice of parameters, and the methods would lead to better guarantees. Seeing the full picture, refining the previous analysis, choosing the right approaches, and carefully handling all details are non-trivial and essential parts of achieving the final result.

---

We hope this comment clarifies that our work is strong and important for the RL community, as it provides the new state-of-the-art complexities compared to the previous approaches, including the paper [1]. Our work is relevant since asynchronous and parallel RL is quickly gaining importance as the availability of GPUs, CPUs, and other computational devices increases. Thank you once again!

Authors

---

[1]: Guangchen Lan, Dong-Jun Han, Abolfazl Hashemi, Vaneet Aggarwal, and Christopher G Brinton. Asynchronous federated reinforcement learning with policy gradient updates: Algorithm design and convergence analysis. (ICLR 2025)

---

### Meta-Review · Area_Chair_VguX · 2025-12-21

**Summary:**

This paper tackles an important and timely problem of efficient distributed and asynchronous policy-gradient reinforcement learning (RL), including heterogeneous worker environments. It proposes two algorithms, i.e., Rennala NIGT for homogeneous and Malenia NIGT for heterogeneous settings. All three reviewers generally agree that this paper’s wall-clock and time-complexity framing and the theoretical convergence guarantees are valuable. A few key discussions include:

- The primary negative reviewer argues that the algorithm and analysis look like a direct combination of asynchronous/federated SGD machinery, and existing NIGT-based policy gradient analyses. The rebuttals addresses this concerns by positioning this work more clearly against existing works. See below for more details.
- Another critique is about the lower bound, and the initial version omitted discussion and comparisons to RL methods achieving improved stationarity rates. Again, the rebuttals made efforts to improve the lower bound.
- One reviewer questions whether a particular heterogeneity construction in the experiments is truly heterogeneous or just a single augmented environment.

**Reviewer Concerns:**

Following about, the key concerns below have been addressed during the rebuttal:
- The authors' rebuttal indicates that they added the previously-missing discussion of methods with improved stationarity rates and expanded related work to include additional federated PG papers, including heterogeneous-environment PG. This directly addresses the reviewers' concerns on the novelty of this work and the missing comparisons with relevant works.
- Regarding the lower bound, the author's rebuttal clarifies that the lower bound targets a general oracle model without extra RL-specific assumptions,. It also explicitly explains why Hessian-aided approaches are not the focus, and positions extending lower bounds under stronger assumptions as future work. This resolves much of the framing confusion.
- Regarding the experimental settings, the authors respond that the construction forces the function distributions to differ and is meant as a controlled toy stress test. They argue that while humans can "see" it as an augmented-state homogeneous environment, the learner must still infer the distinction.

On the other hand, despite that the rebuttal provides a reasonable defense on the experimental settings, if a reviewer views the constructed setting as not representative of real environment heterogeneity, this concern may persist unless additional heterogeneity experiments (more realistic task differences, dynamics changes, reward shifts, different MDPs) are provided. In addition, the positive reviewer notes the upper and lower gap remains open. The authors acknowledge it as an open problem.

**Reviewer Scores:**

Reviewer Codc acknowledged the rebuttal, e.g.m on the added comparisons and clarifications, but state they remain unconvinced on novelty and keep the score unchanged.

Reviewer GmQ8's main concerns are also novelty and the heterogeneity experiment design. The authors' rebuttal addresses both, but given the already cautious stance, the most plausible outcome is no score change.

Reviewer KcK2's flagged weakness is mainly the remaining optimality gap and a question about speed-dependence under heterogeneity. Those issues have been largely explained by the rebuttal. Nothing suggests a downgrade.

---

### Decision · Program_Chairs · 2026-01-26

Accept (Poster)